# Cellular Mechanisms for Antinociception Produced by Oxytocin and Orexins in the Rat Spinal Lamina II—Comparison with Those of Other Endogenous Pain Modulators

**DOI:** 10.3390/ph12030136

**Published:** 2019-09-16

**Authors:** Eiichi Kumamoto

**Affiliations:** Department of Physiology, Saga Medical School, 5-1-1 Nabeshima, Saga 849-8501, Japan; kumamote@cc.saga-u.ac.jp

**Keywords:** oxytocin, orexin A, orexin B, spinal dorsal horn, synaptic transmission, depolarization, antinociception, patch clamp, rat

## Abstract

Much evidence indicates that hypothalamus-derived neuropeptides, oxytocin, orexins A and B, inhibit nociceptive transmission in the rat spinal dorsal horn. In order to unveil cellular mechanisms for this antinociception, the effects of the neuropeptides on synaptic transmission were examined in spinal lamina II neurons that play a crucial role in antinociception produced by various analgesics by using the whole-cell patch-clamp technique and adult rat spinal cord slices. Oxytocin had no effect on glutamatergic excitatory transmission while producing a membrane depolarization, γ-aminobutyric acid (GABA)-ergic and glycinergic spontaneous inhibitory transmission enhancement. On the other hand, orexins A and B produced a membrane depolarization and/or a presynaptic spontaneous excitatory transmission enhancement. Like oxytocin, orexin A enhanced both GABAergic and glycinergic transmission, whereas orexin B facilitated glycinergic but not GABAergic transmission. These inhibitory transmission enhancements were due to action potential production. Oxytocin, orexins A and B activities were mediated by oxytocin, orexin-1 and orexin-2 receptors, respectively. This review article will mention cellular mechanisms for antinociception produced by oxytocin, orexins A and B, and discuss similarity and difference in antinociceptive mechanisms among the hypothalamic neuropeptides and other endogenous pain modulators (opioids, nociceptin, adenosine, adenosine 5’-triphosphate (ATP), noradrenaline, serotonin, dopamine, somatostatin, cannabinoids, galanin, substance P, bradykinin, neuropeptide Y and acetylcholine) exhibiting a change in membrane potential, excitatory or inhibitory transmission in the spinal lamina II neurons.

## 1. Introduction

Somatosensory information from the periphery to spinal cord is transmitted through the dorsal root to the dorsal horn whose gray matter is divided into six laminae named I–VI by Rexed [1]. Nociception among this information is transferred by fine myelinated Aδ and unmyelinated C primary-afferent fibers contained in the dorsal root to neurons in the superficial laminae of the dorsal horn, particularly lamina II (substantia gelatinosa), in a mono- and polysynaptic manner ([2,3]; see [4] for review). This nociceptive information flows to the thalamus through a connection with projection neurons in lamina I and deeper laminae of the spinal dorsal horn (see [5] for review), and then to the primary sensory area of the cerebral cortex, producing nociceptive sensation. Since the proposal of the gate-control theory of pain by Melzack and Wall [6], a modulation of synaptic transmission in spinal lamina II neurons is thought to play a crucial role in the regulation of nociceptive sensation (see [4,7] for review). This synaptic modulation mainly occurs at pre- and/or postsynaptic sites of excitatory and inhibitory synapses in spinal lamina II neurons through actions of a number of endogenous substances that are either locally produced/released in the spinal dorsal horn or released from descending neurons originating from higher centers such as the brainstem and hypothalamus.

Consistent with the importance of spinal lamina II neurons, a plastic change in glutamatergic primary-afferent inputs to the neurons occurs in hyperalgesia caused by an intraplantar injection of complete Freund’s adjuvant [8] or ovariectomy [9] in rats. In partial nerve injury rat models compared with sham rats, primary-afferent-evoked inhibitory transmission is inhibited in spinal lamina II neurons; γ-aminobutyric acid (GABA)-synthesizing enzyme expression is reduced in level in the spinal dorsal horn [10]. There is a difference between wild-type and *Glra3^-/-^* mice in mechanical and thermal behavior of pain model produced by complete Freund’s adjuvant injection, indicating an involvement of glycine-receptor α3 subunit (see [11] for review) in inflammatory pain [12]. A decrease in expression of the potassium-chloride exporter KCC2 shifts transmembrane chloride gradient and thus causes normally inhibitory anionic synaptic currents to be excitatory in rat spinal dorsal horn neurons; as a result, nociceptive thresholds are markedly reduced ([13]; see [14] for review). Recently, Medrano et al. [15] have suggested that a shift in the reversal potential for chloride is an important component of a loss of inhibitory tone in neuropathic pain mouse models produced by nerve injury.

Many of the targets on which the endogenous analgesics act are membrane proteins including receptors for neurotransmitters. There are many investigations about the effects of the analgesics on synaptic transmission in lamina II neurons by using the whole-cell patch-clamp technique in spinal cord slice preparations dissected from 2- to 8-week-old rodents. Lamina II layer is easily discernable as a translucent band in slices under a binocular microscope with light transmitted from below ([16,17]; see [18,19] for review). Recently, hypothalamus-derived neuropeptides, oxytocin, orexins A and B, that have an ability to alleviate pain when intrathecally administrated, were shown to exhibit a similar synaptic modulation, i.e., inhibitory transmission facilitation following excitatory transmission facilitation, in adult rat spinal lamina II neurons [20,21,22]. Their modulatory actions were either similar to or different from those of the other endogenous pain modulators (opioids, nociceptin, adenosine, adenosine 5’-triphosphate (ATP), noradrenaline, serotonin, dopamine, somatostatin, cannabinoids, galanin, substance P, bradykinin, neuropeptide Y and acetylcholine) in the spinal cord level. These modulators produce a membrane depolarization or hyperpolarization, a facilitation or depression of excitatory transmission, or a facilitation or depression of inhibitory transmission. This review article will mention cellular mechanisms for synaptic modulation produced by oxytocin, orexins A and B and also other endogenous pain modulators’ ones that are related to the present topics, and then discuss a similarity and difference in antinociceptive mechanisms among these modulators.

## 2. Fast Synaptic Transmission in Spinal Lamina II Neurons

As in the brain, excitatory transmission in the lamina II is mediated by L-glutamate released from nerve terminals; this amino acid originates from primary-afferent fibers and glutamatergic interneurons in the lamina II [17]. On the other hand, inhibitory transmission in lamina II neurons is mediated not only by GABA as in the brain but also by glycine; these amino acids are released from the terminals of GABAergic and/or glycinergic neurons that exist in the spinal dorsal horn or that originate from supraspinal regions such as the rostral ventromedial medulla [23,24,25].

Although fast excitatory transmission in the central nervous system is generally mediated by ionotropic α-amino-3-hydroxy-5-methyl-4-isoxazole propionate (AMPA) and *N*-methyl-D-aspartate (NMDA) receptor-channels (see [11] for review), spontaneous and electrically-evoked excitatory postsynaptic currents (EPSCs) recorded at –70 mV in lamina II neurons are due to the activation of AMPA receptor-channels, because they are completely blocked by an AMPA receptor-channel antagonist 6-cyano-7-nitroquinoxaline-2,3-dione (CNQX, 10 μM; for example, see [20,26]). NMDA receptor-channels are activated at more positive potentials than –70 mV (see Section 4.11 and Section 4.12). On the other hand, inhibitory postsynaptic currents (IPSCs) are mediated by ionotropic GABA_A_ and glycine receptor-channels ([23]; see [11] for review of the receptor-channels).

L-Glutamate released from the central terminals of primary-afferent Aδ-fibers or C-fibers to lamina II neurons can be examined by measuring monosynaptic EPSCs evoked by stimulating the dorsal root. Aδ-fiber or C-fiber-evoked EPSCs are distinguished on the basis of the conduction velocity of afferent fibers and stimulus threshold to elicit EPSCs; the Aδ-fiber or C-fiber responses, respectively, are considered as monosynaptic in origin when the latency remains constant and there is no failure during repetitive stimulation at 20 Hz for 1 sec, or when failures do not occur during repetitive stimulation at 2 Hz for 10 sec (for example, see [8,27]). On the other hand, L-glutamate released spontaneously from primary-afferent central terminals and glutamatergic interneuron terminals to lamina II neurons can be examined by measuring spontaneous EPSCs (sEPSCs). sEPSCs in lamina II neurons in adult rat spinal cord slices were unaffected in frequency and amplitude by the voltage-gated Na^+^-channel blocker tetrodotoxin (TTX, 0.5 μM; for example, see [20]), probably because of deafferentiation in the slices used. This result indicates that all of sEPSCs occur without the propagation of action potentials from cell soma, whose neuron is presynaptic to lamina II neurons, to the terminals, resulting in spontaneous releases under our experimental conditions.

GABAergic and glycinergic spontaneous IPSCs (sIPSCs) are, respectively, recorded in the presence of the glycine-receptor antagonist strychnine and the GABA_A_-receptor antagonist bicuculline. GABAergic sIPSCs are about three-fold longer in duration than glycinergic ones (for example, see [28]). GABAergic and glycinergic sIPSCs were unaffected in frequency and amplitude by TTX (0.5 μM) or a mixture of CNQX (10 μM) and an NMDA receptor-channel antagonist DL-2-amino-5-phosphonovaleric acid (APV, 50 μM; [28]). This result indicates that sIPSCs occur in a manner independent of action potential production, and AMPA and NMDA receptor activation under our experimental conditions.

## 3. Effects of Hypothalamus-Derived Neuropeptides on Synaptic Transmission in Spinal Lamina II Neurons

### 3.1. Oxytocin Action

A posterior pituitary hormone oxytocin (a 9-amino acid peptide) has various actions including social interaction and antinociception, other than milk ejection during lactation and uterine contraction during parturition [29,30,31,32]. There is much evidence showing that oxytocin plays a role in regulating nociceptive transmission to the spinal dorsal horn from the periphery. First, there are oxytocin-immunoreactive fibers to the spinal superficial dorsal horn (SDH) from the hypothalamic paraventricular nucleus [33,34], and oxytocinergic axons make synaptic contacts with spinal SDH neurons [35]. Second, oxytocin-binding sites densely exist in the spinal dorsal horn [36,37,38,39,40]. Third, the electrical stimulation of the anterior part of the hypothalamic paraventricular nucleus increased oxytocin concentration in cerebrospinal fluid and produced antinociception [41]. Somatic noxious stimulation activated hypothalamic paraventricular oxytocinergic neurons projecting to the spinal dorsal horn [42]. Intraperitoneal or intrathecal administration of oxytocin reportedly produced antinociception in rats [43,44].

#### 3.1.1. Action of Oxytocin on Holding Current

In 67% of the adult (6–8 weeks old) rat lamina II neurons tested, oxytocin (0.5 μM) produced an inward current at –70 mV (membrane depolarization); the peak amplitude of this current averaged to be 12.6 pA. Remaining all neurons except for neurons (1%) exhibiting an outward current had no effect on holding currents. The peak inward current was concentration-dependent with a half-maximal effective concentration (EC_50_) value of 0.022 μM. This activity of oxytocin was slow in recovery after its washout. This desensitization may be due to an internalization of oxytocin receptors [29]. Oxytocin-induced inward current was resistant to TTX (0.5 μM), indicating no involvement of action potential production [20]. Moreover, the inward current persisted in Krebs solution containing CNQX (10 μM) and nominally Ca^2+^-free Krebs solution, indicating a direct action of oxytocin and also no involvement of Ca^2+^ entry from extracellular solution.

Oxytocin current was mimicked by an oxytocin-receptor agonist TGOT ([Thr^4^,Gly^7^]-oxytocin; 0.5 μM) and was reduced in amplitude by an oxytocin-receptor antagonist dVOT ([d(CH_2_)_5_^1^,Tyr(Me)^2^,Thr^4^,Orn^8^,des-Gly-NH_2_^9^]-vasotocin; 1 μM), indicating an involvement of oxytocin receptors. This result is consistent with the localization of oxytocin receptors in the adult rodent spinal SDH [45,46,47]. Consistent with the fact that oxytocin receptors are coupled to guanosine 5’-triphosphate (GTP)-binding proteins (G proteins), oxytocin activity disappeared 60 min after the whole-cell configuration using patch-pipette solutions containing guanosine 5'-*O*-(2-thiodiphosphate) (GDP-β-S, 1 mM; which has an ability to block the actions of G proteins) [20].

Oxytocin receptors trigger G_q/11_, G_s_ or G_i/o_ protein-mediated cellular signaling cascades [32]. Oxytocin-induced inward current was decreased in amplitude by U-73122 (10 μM; an inhibitor of phospholipase C (PLC) coupled to G_q/11_ protein [48]), indicating an involvement of PLC. Phosphatidylinositol 4,5-bisphosphate hydrolysis caused by PLC leads to the production of two second messengers, diacylglycerol (DAG) that activates protein kinase C (PKC) and inositol 1,4,5-triphosphate (IP_3_) that releases Ca^2+^ from intracellular Ca^2+^ stores. Oxytocin activity was suppressed by 2-aminoethoxydiphenyl borate (200 μM; an IP_3_-induced Ca^2+^-release inhibitor [49]) but not dantrolene (10 μM; a Ca^2+^-induced Ca^2+^-release inhibitor [50]), chelerythrine (10 μM; a PKC inhibitor [51]) and dibutyryl cyclic-AMP (1 mM), a membrane-permeable analogue of cyclic-AMP (whose intracellular concentration is regulated by G_s_ and G_i/o_ protein). Current–voltage relation for the oxytocin current reversed at negative potentials more than the equilibrium potential for K^+^ or around 0 mV. The oxytocin current was decreased in amplitude in high-K^+^ (10 mM), low-Na^+^ (decreased by 117 mM) or Ba^2+^ (1 mM)-containing Krebs solution. These results indicate that the inward current is due to an alteration in membrane permeabilities to K^+^ and/or Na^+^, which is possibly mediated by PLC (whose activation is due to α subunit of G_q/11_ protein) and IP_3_-induced Ca^2+^ release [20]. The oxytocin current may be mediated by closed K^+^ channels, as seen in muscarine-sensitive K^+^ channels [52], and opened Na^+^ channels; the former idea is supported by a sensitivity of oxytocin current to Ba^2+^. Breton et al. [53] have reported a closure of A-type and delayed-rectifier K^+^-channels produced by TGOT in young rat lamina II neurons. A possibility cannot be ruled out in lamina II neurons that not only G_q/11_ but also G_s_ or G_i/o_ protein is activated by oxytocin and thus an interaction among activations of their G proteins occurs (see [32]).

Oxytocin is known to activate not only oxytocin receptors but also vasopressin receptors with a lower efficacy [54]. Vasopressin receptors are classified into the V_1A_, V_1B_ and V_2_ subtypes; antinociception produced by the systemic administration of oxytocin in the mouse is reportedly mediated by vasopressin V_1A_ receptors [39]. Although a vasopressin V_1A_-receptor antagonist (d(CH_2_)_5_^1^,Tyr(Me)^2^, Arg^8^)-vasopressin (1 μM) diminished the peak amplitude of oxytocin (0.5 μM) current in lamina II neurons, there was no correlation in amplitude between responses of a vasopressin-receptor agonist [Arg^8^]-vasopressin (0.5 μM) and oxytocin (0.5 μM) [20]. Therefore, the lamina II oxytocin response appeared to be not mediated by vasopressin V_1A_ receptors. This idea corresponds to the observation that the labeling of spinal SDH layers by [^125^I]vasopressin antagonist is weaker in extent than that of [^125^I]oxytocin antagonist in the adult rat [46].

In lamina II neurons where oxytocin (0.5 μM) produced an inward current, noradrenaline (20 μM), serotonin (40 μM) or adenosine (1 mM) elicited an outward current while (-)-nicotine (100 μM) or carbamoylcholine (10 μM) produced an inward current at –70 mV ([20]; see below).

#### 3.1.2. Action of Oxytocin on Excitatory Transmission

Oxytocin (0.5 μM) did not affect the frequency and amplitude of sEPSC, and monosynaptically-evoked primary-afferent Aδ-fiber and C-fiber EPSC amplitude in adult rat lamina II neurons [20]. On the other hand, TGOT is reported to increase the spontaneous release of L-glutamate from nerve terminals in spinal SDH neurons of young (2–4 weeks old) rats [55]. This difference between adult and young rats appeared to be due to a developmental change in synaptic modulation produced by oxytocin. Indeed, in young (11-21 postnatal days) rats, oxytocin (0.5 μM) produced not only an inward or outward current at –70 mV but also presynaptically inhibited or facilitated spontaneous excitatory transmission, depending on the neurons tested [56]. It has been also reported that there is a difference between adult male and female rats in synaptic modulation produced by oxytocin (0.5 μM) in lamina II neurons [56].

Although an oxytocin-induced L-glutamate release increase has been reported in neonate rat spinal SDH neurons in culture [36], Robinson et al. [57] have found an inhibition by oxytocin of primary-afferent-evoked excitatory transmission in adult mouse spinal SDH neurons. There seems to be a difference in excitatory transmission modulation produced by oxytocin among distinct animal species and also ages.

#### 3.1.3. Action of Oxytocin on Inhibitory Transmission

Oxytocin (0.5 μM) increased the frequency of GABAergic and glycinergic sIPSC with a small increase in its amplitude in >90% of the adult rat lamina II neurons tested. The extents of the GABAergic sIPSC frequency and amplitude increase averaged 438% and 66%, respectively, and those of the glycinergic ones 578% and 35%, respectively. These facilitatory effects were slow in recovery after washout of oxytocin, as seen in its depolarized effect. The activity of oxytocin was concentration-dependent; EC_50_ values for it to increase GABAergic and glycinergic sIPSC frequency were 0.024 and 0.038 μM, respectively. These activities were mimicked by TGOT (0.5 μM), depressed by dVOT (1 μM) and TTX (0.5 μM), indicating an involvement of oxytocin receptors and action potential production [20]. Breton et al. [55] also reported spontaneous GABAergic transmission enhancement produced by TGOT in young rat spinal SDH neurons, although glycinergic transmission was not examined.

These results indicate that oxytocin produces a membrane depolarization in lamina II neurons by activating oxytocin receptors, which increases the neuronal activity of the neurons, leading to the enhancement of inhibitory transmission, a possible mechanism for antinociception [20]. This idea is supported by the observation that the EC_50_ values (0.024-0.038 μM) for sIPSC frequency increase are closed to that (0.022 μM) for depolarization production. On the other hand, Breton et al. [55] have proposed the idea that an increase in the spontaneous release of L-glutamate onto GABAergic neurons, produced by oxytocin-receptor activation, results in GABAergic transmission enhancement, leading to antinociception.

Recently, oxytocin has been shown to increase intracellular Ca^2+^ concentration and hyperpolarize membranes in cultured rat capsaicin-sensitive dorsal root ganglion (DRG) neurons [58]. Moreover, oxytocin is reported to activate transient receptor potential vanilloid-1 (TRPV1) channels [59]. TRPV1 channel activation produced by capsaicin results in sEPSC frequency increase and monosynaptically-evoked primary-afferent C-fiber EPSC amplitude reduction in lamina II neurons ([60,61]; for review, see [62]). Since oxytocin (0.5 μM) did not affect sEPSC frequency and C-fiber EPSC amplitude (see above), oxytocin at this concentration did not appear to activate TRPV1 channels located in primary-afferent central terminals in the lamina II.

### 3.2. Orexins Action

There is much evidence demonstrating that hypothalamic neuropeptides, orexin A (hypocretin 1, 33-amino acid peptide) and orexin B (hypocretin 2, 28-amino acid peptide; [63,64]), play a pivotal role in not only arousal/wakefulness (for review, see [65,66,67]) but also in inhibiting nociceptive transmission in the spinal dorsal horn. For instance, orexinergic fibers in the hypothalamus project to the spinal dorsal horn in rodents [68], orexins A and B exist in the rat spinal cord, albeit the latter is more highly located than the former [69], and the rat spinal cord expresses G protein-coupled orexin-1 and orexin-2 receptors [70,71] that are activated by orexins A and B [64]. Orexin-1 receptors bind more effectively (by about 100 fold) orexin A than orexin B while orexin-2 receptors bind orexins A and B with a comparable affinity [64]. Intrathecal administration of orexins A and B produces antinociception in rodents [72,73,74].

#### 3.2.1. Action of Orexin A on Holding Current

In 18% of the adult rat lamina II neurons examined, orexin A (0.05 μM) produced an inward current at –70 mV (membrane depolarization) that was not accompanied by a change in spontaneous excitatory transmission. On the other hand, 19% of the lamina II neurons did not alter holding currents while exhibiting an increase in sEPSC frequency. Both of these orexin A actions were elicited in 50% of the lamina II neurons. Remaining neurons (13%) did not respond to orexin A. This variability possibly results from a heterogeneity in orexin A receptor expression among different neurons in the lamina II such as islet, central, medial-lateral, radial or vertical type neurons [75]. The peak amplitude of the inward current averaged to be 5.9 pA. The inward current production was repeated, and was concentration-dependent with an EC_50_ value of 0.0045 μM [21].

The inward current produced by orexin A was reduced in amplitude by an orexin-1 receptor antagonist SB334867 (*N*-(2-methyl-6-benzoxazolyl)-*N*'-1,5-naphthyridin-4-yl urea; 1 μM) but not an orexin-2 receptor antagonist JNJ10397049 (*N*-(2,4-dibromophenyl)-*N*'-[(4*S*,5*S*)-2,2-dimethyl-4 -phenyl-1,3-dioxan-5-yl]-urea; 1 μM; see [76] for their antagonists) and TTX (0.5 μM), indicating an involvement of a direct activation of orexin-1 but not orexin-2 receptors by orexin A without action potential production [21].

#### 3.2.2. Action of Orexin A on Excitatory Transmission

The sEPSC frequency increase produced by orexin A (0.05 μM) averaged 54% in extent, and was not accompanied by a change in sEPSC amplitude. The sEPSC frequency increase was repeated and was concentration-dependent with an EC_50_ value of 0.030 μM [21]. The sEPSC frequency increase produced by orexin A was diminished in extent by SB334867 (1 μM) but not JNJ10397049 (1 μM) and TTX (0.5 μM), indicating an involvement of a direct activation of orexin-1 but not orexin-2 receptors by orexin A without action potential production [21].

Jeon et al. [77] have reported a similar inward current and spontaneous excitatory transmission enhancement produced by orexin A in young rat lamina II neurons.

#### 3.2.3. Action of Orexin A on Inhibitory Transmission

In 51% of the adult rat lamina II neurons examined, orexin A (0.05 μM) enhanced GABAergic spontaneous transmission that was observed in the presence of strychnine (1 μM). The extents of sIPSC frequency and amplitude increases averaged 119% and 33%, respectively. The orexin A activity was repeated. The GABAergic transmission enhancement was suppressed in extent by SB334867 (1 μM) and TTX (0.5 μM) but not JNJ10397049 (1 μM), indicating an involvement of an activation of orexin-1 but not orexin-2 receptors by orexin A and action potential production [21].

In 79% of the lamina II neurons examined, orexin A (0.05 μM) enhanced glycinergic spontaneous transmission that was observed in the presence of bicuculline (20 μM). The extents of sIPSC frequency and amplitude increases averaged 85% and 38%, respectively. The glycinergic transmission enhancement was decreased in extent by SB334867 (1 μM) and TTX (0.5 μM) but not JNJ10397049 (1 μM), indicating an involvement of an activation of orexin-1 but not orexin-2 receptors by orexin A and action potential production [21].

The GABAergic and glycinergic transmission enhancements are suggested to occur as a result of the production of action potentials by membrane depolarization and increased L-glutamate release through orexin-1 receptors activated by orexin A [21].

### 3.3. Orexin B Action

#### 3.3.1. Action of Orexin B on Holding Current

As with orexin A, orexin B (0.05 μM) produced an inward current at –70 mV (membrane depolarization) and/or sEPSC frequency increase in adult rat lamina II neurons. In 16% of the neurons tested, orexin B produced an inward current with no change in spontaneous excitatory transmission. On the other hand, 18% of the lamina II neurons did not alter holding currents while producing sEPSC frequency increase. Both of these orexin B actions were elicited in 32% of the lamina II neurons. Remaining neurons (34%) did not respond to orexin B. Although such a variability in response is possibly due to a heterogeneity of lamina II neurons expressing orexin B receptors [75], it has been reported in the young rat that lamina II neurons exhibiting orexin B activity are greater in proportion in radial or vertical neurons than central or unclassified neurons [78]. The peak amplitude of the inward current in adult rats averaged to be 6.5 pA. The inward current production was repeated and was concentration-dependent with an EC_50_ value of 0.020 μM [22]. This value was four-fold larger than that of orexin A while being comparable to that (0.022 μM) for oxytocin to produce an inward current (see above). Although ionic mechanisms for the inward current produced by orexin B could not be investigated due to its small amplitude, it was suggested that orexin B-induced inward current in young rats may result from a change in membrane permeability to multiple ions [78]. Orexins-induced inward current has been generally attributed to inhibition of K^+^ channels, stimulation of Na^+^/Ca^2+^-exchanger and/or activation of non-selective cation channels (see [79] for review).

The inward current produced by orexin B was resistant to SB334867 (1 μM) and TTX (0.5 μM) while being sensitive to JNJ10397049 (1 μM), indicating an involvement of a direct activation of orexin-2 but not orexin-1 receptors by orexin B without action potential production [22].

#### 3.3.2. Action of Orexin B on Excitatory Transmission

The sEPSC frequency increase produced by orexin B (0.05 μM) averaged 71% in extent, and was not accompanied by a change in sEPSC amplitude. As seen in orexin A activity, the sEPSC frequency increase was repeated and was concentration-dependent with an EC_50_ value of 0.039 μM [22]. This value was almost comparable to that of orexin A (see above).

The sEPSC frequency increase produced by orexin A was suppressed in extent by JNJ10397049 (1 μM) but not SB334867 (1 μM) and TTX (0.5 μM), indicating an involvement of a direct activation of orexin-2 but not orexin-1 receptors by orexin B without action potential production [22]. This result is consistent with the observations that orexin B is much more potent for orexin-2 than orexin-1 receptors [64] and that orexin-2 receptors exist at high densities in the rat spinal SDH [70].

Grudt et al. [78] have reported that orexin B increases sEPSC frequency with no change in dorsal root-evoked EPSC amplitudes in young rat spinal lamina II neurons.

#### 3.3.3. Action of Orexin B on Inhibitory Transmission

In 71% of the adult rat lamina II neurons examined, orexin B (0.05 μM) enhanced glycinergic spontaneous transmission that was observed in the presence of bicuculline (20 μM). The extents of sIPSC frequency and amplitude increases averaged 110% and 54%, respectively. The orexin B activity was repeated. The glycinergic transmission enhancement was reduced in extent by JNJ10397049 (1 μM) and TTX (0.5 μM) but not SB334867 (1 μM), indicating an involvement of an activation of orexin-2 but not orexin-1 receptors by orexin B and action potential production [22]. It is suggested that the glycinergic transmission enhancement may occur as a result of the production of action potentials by membrane depolarization and increased L-glutamate release through orexin-2 receptors activated by orexin B.

On the other hand, in 24 out of the 26 lamina II neurons tested, orexin B (0.05 μM) had no effect on GABAergic spontaneous transmission that was observed in the presence of strychnine (1 μM). The remaining two neurons exhibited a spontaneous GABAergic transmission enhancement. This result was quite different from that of orexin A (see above; [22]). The result that orexin B enhances glycinergic but not GABAergic transmission is consistent with the observation that strychnine but not bicuculline depresses enhanced inhibitory transmission produced by orexin B in young rats [78]. This difference in orexin B activity between GABAergic and glycinergic transmission was distinct from the activity of oxytocin that enhanced both GABAergic and glycinergic transmission (see above).

In lamina II neurons sensitive to orexin B (0.05 μM), orexin A (0.05 μM) produced an inward current at –70 mV and sEPSC frequency increase with extents comparable to those of orexin B. Moreover, orexin A (0.05 μM) increased glycinergic sIPSC frequency and amplitude with similar extents to those of orexin B (0.05 μM) in the same neuron [22]. This observation that individual lamina II neurons are sensitive to both orexins A and B may be consistent with the fact that both orexins A and B are produced from a precursor peptide and that orexins A and B co-localize in the cat hypothalamus and brain stem [80].

As shown above, there was a difference between orexins A and B in synaptic modulation in lamina II neurons. Orexin A produced a membrane depolarization about four-fold more effectively than orexin B; orexin A enhanced both GABAergic and glycinergic transmission while orexin B only glycinergic transmission. These results may explain a difference between orexins A and B in antinociceptive effects, that is, more effectiveness of orexin A than orexin B [72,73].

Orexin B-sensitive lamina II neurons responded to oxytocin (0.5 μM) with the production of an inward current at –70 mV, but did not exhibit any change in spontaneous excitatory transmission following oxytocin application, indicating clearly a difference in synaptic modulation between orexins and oxytocin [22].

Although orexin B activities similar to those of our studies were reported in young rats [78], orexin B-responsive neurons appeared to be larger in proportion in adult (66–71%) than young (12–29%) rats, albeit orexin B was used at a higher concentration (1 μM) in young rats. There appeared to be a developmental change in orexin B activities.

Post- and presynaptic orexins’ actions similar to those in rat spinal lamina II neurons have been found in other preparations, such as rodent histaminergic tuberomammillary [81], laterodorsal tegmental [82], median preoptic nucleus [83], pedunculopontine tegmental [84], rostral ventrolateral medulla [85] and orexin neurons [86].

## 4. Effects of Other Endogenous Pain Modulators on Synaptic Transmission in Spinal Lamina II Neurons

### 4.1. Opioid Actions

Opioids activate three subtypes of opioid receptors, μ-, δ- and κ-type, leading to voltage-gated Ca^2+^ channel inhibition, inwardly-rectifying K^+^ channel activation (both of which are due to βγ subunit) or adenylate cyclase inhibition (which is mediated by α subunit) through G_i/o_ protein activation (for review, see [87]). These opioid receptors are expressed in the spinal SDH, especially lamina II in rats ([88,89]; for review, see [90]). Opioid-binding sites were partially reduced in number in the dorsal horn after the disruption of primary afferents by dorsal rhizotomy [88] or the pretreatment with capsaicin [91], indicating the localization of opioid receptors in the dorsal horn. The rat lamina II reportedly contains endogenous opioid peptides such as enkephalins [92,93], endomorphin-1 (Tyr-Pro-Trp-Phe-NH_2_) and endomorphin-2 (Tyr-Pro-Phe-Phe-NH_2_; which is different by only one amino acid from endomorphin-1) [94,95], the latter two of which bind to the μ-opioid receptor with a high affinity compared to δ- and κ-opioid receptors (for review, see [96]). Furthermore, endomorphin-2-like substances were released from the rat spinal dorsal horn in response to electrical stimulation applied to the dorsal root entry zone [97]. Intrathecal administration of opioids produced a powerful analgesia in rats ([98]; for review, see [99]). Opioids administrated into the lamina II in anesthetized cats suppressed an excitation of deeper dorsal horn neurons caused by noxious peripheral stimuli without a change in their responses to innocuous stimuli such as touch [100].

μ- and δ-Type opioid-receptor agonists [(D-Ala^2^, *N*-Me-Phe^4^, Gly^5^-ol)enkephalin (DAMGO) and (D-Pen^2^, D-Pen^5^)enkephalin (DPDPE), respectively; each 1 μM] reduced the peak amplitude of monosynaptically-evoked Aδ-fiber EPSC and also the frequency of miniature EPSC (mEPSC) recorded in the presence of TTX without a change in its amplitude in 70–100% of the adult rat lamina II neurons tested. The actions of DAMGO and DPDPE were not seen in the presence of μ- and δ-opioid receptor antagonists [D-Phe-Cys-Tyr-D-Trp-Arg-Thr-Pen-Thr-NH_2_ (CTAP) and naltrindole, respectively; each 1 μM], respectively, indicating the presence of μ- and δ-type opioid receptors involved in inhibiting the release of L-glutamate from primary-afferent central terminals and from interneuron terminals [101]. Monosynaptic C-fiber-evoked EPSC was decreased in peak amplitude by DAMGO more effectively than monosynaptic Aδ-fiber-evoked one [102]. On the other hand, κ-type opioid-receptor agonist D-(5 α,7 α,8 β)-(+)-*N*-methyl-*N*-[7-(1-pyrrolidinyl)-] -oxaspiro[4,5] dec-8-yl]benzeneacetamide (U-69593; 1 μM) reduced monosynaptic Aδ-fiber EPSC amplitude and mEPSC frequency in only 30% of the lamina II neurons tested [101]. A similar inhibitory effect of U-69593 (0.3 μM) on mEPSC frequency has been demonstrated in young rat lamina II neurons [103]. In about 50% of the adult rat lamina II neurons examined, endomorphin-1 and endomorphine-2 (each 1 μM) activated inwardly-rectifying K^+^ channels, resulting in an outward current at –70 mV (membrane hyperpolarization). Such a paucity of responsive neurons may be due to a heterogeneity in μ-opioid receptor expression among different lamina II neurons [75]. These endomorphin actions were concentration-dependent with almost the same EC_50_ value (0.19-0.21 μM) and inhibited by CTAP (1 μM), indicating the activation of μ-opioid receptors in postsynaptic neurons ([104]; for review, see [105]). Yajiri and Huang [106] reported an inhibition of primary-afferent Aδ-fiber-evoked excitatory transmission by endomorphin-1 or endomorphin-2, although their actions were not compared in extent with each other. Endogenous opioids other than endomorphins are reported to hyperpolarize membranes [107]. Although not only μ- but also δ- and κ-opioid receptor agonists exhibited a membrane hyperpolarizing effect in rat lamina II neurons, this effect in individual neurons was distinct in responsiveness among these agonists; μ agonist was effective in a higher proportion of the lamina II neurons tested compared to other agonists [108]. Regarding inhibitory transmission, spontaneous and focally-evoked transmissions mediated by GABA and glycine were not affected by DAMGO (1 μM), where the focal stimulation was performed by using an electrode put within 150 μm of the neuron recorded [101]. In young rat lamina II neurons, both endomorphin-1 and endomorphin-2 are reported to hyperpolarize membranes [94] and to inhibit excitatory transmission [94,109]. Kerchner and Zhuo [110] have reported that DAMGO (1 μM) reduces evoked IPSC amplitude and the frequency of miniature IPSC (mIPSC) recorded in the presence of TTX in rat dorsal horn neurons; it is unknown why there is a difference between their results and ours.

### 4.2. Nociceptin Action

Nociceptin (a 17-amino acid peptide), which is also known as orphanin FQ, activates G protein-coupled opioid receptor-like1 (ORL1) receptors (recently called nociceptin opioid peptide receptors, NOP receptors) that are similar in structure to the opioid receptors but do not bind opioids. The activation of the NOP receptors results in an inhibition of voltage-gated Ca^2+^ channels, an activation of inwardly-rectifying K^+^ channels (both of which by βγ subunit) or an inhibition of adenylate cyclase (by α subunit) by activating G_i/o_ protein, as seen in opioid receptor activation [111,112]. In spite of such a similarity between the NOP and opioid receptors, these receptors exhibit remarkable differences in receptor functions such as phosphorylation, desensitization and internalization and in their modulatory functions of nociceptive transmission (see [113,114] for review). NOP receptors are densely distributed in the SDH of the adult rat spinal cord [115]. Nociceptin peptide and pre-pronociceptin mRNA [116,117] are densely distributed in the SDH of the rodent spinal cord. Both nociceptin and NOP receptor immunoreactivities were upregulated in rat DRG neurons after nerve injury and inflammation [118]. Furthermore, nociceptin-like substances are reported to be released from the rat spinal dorsal horn in response to electrical stimulation applied to the dorsal root entry zone [119]. According to behavioral studies in adult rats, intrathecal administration of nociceptin produced an antinociceptive effect [120,121,122] and attenuated hyperalgesia in a model of sciatic-nerve injury [123] as well as of inflammation by carageenan injection [124].

In adult rat lamina II neurons, nociceptin (1 μM) activated an inwardly-rectifying K^+^ channel, resulting in membrane hyperpolarization; this activity was concentration-dependent with an EC_50_ value of 0.23 μM [125]. Nociceptin (1 μM) also reduced the peak amplitudes of monosynaptic Aδ-fiber and C-fiber EPSCs evoked in lamina II neurons by stimulating the dorsal root, where C-fiber EPSCs (57% peak amplitude reduction) were more sensitive to nociceptin than Aδ-fiber ones (30%). Since nociceptin did not affect a response of lamina II neurons to bath-applied AMPA, this action was presynaptic in origin, i.e., due to a decrease in the release of L-glutamate from primary-afferent central terminals [126]. These nociceptin actions were inhibited by a nociceptin precursor nocistatin (1 μM) and an NOP receptor antagonist, 1-[(3*R*, 4*R*)-1-cyclooctylmethyl-3-hydroxymethyl-4-piperidyl]-3-ethyl-1,3-dihydro-2*H*-benzimidazol-2-one (CompB or J-113397; 1-3 μM [127]), indicating that the activation of NOP receptor leads to a decrease in the excitability of lamina II neurons and thus to antinociception. Like DAMGO, nociceptin (1 μM) did not affect spontaneous and focally-evoked inhibitory transmissions mediated by GABA and glycine in lamina II neurons [126]. Excitatory transmission inhibition, the lack of changes in inhibitory transmission and membrane hyperpolarization produced by nociceptin have been demonstrated in young rat lamina II neurons [128,129,130].

Although nociceptin exhibits modulatory actions similar to those of opioids in lamina II neurons, nociceptin but not opioids inhibit T-type voltage-gated Ca^2+^ channels in rat DRG neurons, suggesting a difference between the two neuropeptides in modulating primary-afferent-evoked glutamatergic transmission (see [131] for review).

### 4.3. Adenosine Action

Adenosine activates three subtypes of G protein-coupled metabotropic receptors as classified into A_1_-, A_2_- (A_2A_-, A_2B_-) and A_3_-types. The A_1_ and A_2_ adenosine receptors (G_i/o_- and G_s_-protein coupled ones, respectively) are primarily coupled to adenylate cyclase in a negative and positive manner, respectively, while the A_3_ one activates an IP_3_/DAG system through PLC (by α subunit of G_q/11_ protein); the A_1_ receptor also opens K^+^ channels or closes Ca^2+^ channels (by α subunit) through G_i/o_ protein activation (for review, see [132]). The presence of A_1_ adenosine receptors at a high density in the spinal dorsal horn has been demonstrated in terms of A_1_ agonist-binding sites [133], A_1_ adenosine receptor mRNAs [134] and proteins [135]. Adenosine in the lamina II would be either released from neurons and/or glial cells or produced as a result of a cleavage by ectonucleotidases of ATP released from them. In support of a role of adenosine in the lamina II, this region has a high density of the rat equilibrative nucleoside transporter (rENT1) that controls the extracellular level of nucleosides such as adenosine [135]. Behavioral studies have demonstrated that an intrathecal administration of adenosine analogues produces antinociception in the hot plate and tail flick tests [136]. The antinociceptive effect of adenosine has been reported to be due to the activation of the A_1_ adenosine receptor ([137,138]; for review, see [139]). Recent studies have demonstrated an effectiveness of adenosine A_3_ receptor activation in alleviating neuropathic pain in rodents [140]; this antinociceptive effect appears to be due to an inhibition of enhanced microglial activation in the spinal dorsal horn [141].

In adult rat lamina II neurons, adenosine concentration-dependently produced an outward current at –70 mV (membrane hyperpolarization) and reduced the frequency of sEPSC without a change in its amplitude; their EC_50_ values were 177 and 277 μM, respectively [142,143]. The outward current was due to the activation of K^+^ channels that were sensitive to Ba^2+^ (100 μM) and 4-aminopyridine (5 mM) while being resistant to tetraethylammonium (TEA; 5 mM; [143]). Monosynaptic Aδ-fiber and C-fiber-evoked EPSCs were depressed in amplitude by adenosine (100 μM) with a comparable extent (26% and 27%, respectively, in the same neuron). EC_50_ value for adenosine in reducing Aδ-fiber EPSC peak amplitudes was 217 μM, a value similar to those of outward current and sEPSC frequency reduction produced by adenosine [144]. All of the adenosine actions were mimicked by an A_1_ adenosine-receptor agonist, *N*^6^-cyclopentyladenosine (1 μM), and blocked by an A_1_ antagonist, 8-cyclopentyl-1,3-dipropylxanthine (1 μM), indicating an involvement of the A_1_ adenosine receptor [142,143]. Spontaneous and focally-evoked GABAergic and glycinergic IPSCs were also suppressed in frequency and amplitude, respectively, by adenosine (100 μM) through the activation of presynaptic A_1_ adenosine receptors. EC_50_ values for adenosine in reducing evoked GABAergic and glycinergic IPSC amplitude were 14.5 and 19.1 μM, respectively ([145]; see [146] for review). These findings are consistent with the presence of the A_1_ adenosine receptor in the spinal dorsal horn. It may be of interest to note an enhancement of adenosine A_1_ receptor sensitivity at excitatory but not inhibitory synapses in the lamina II in a rat partial nerve-injury model of neuropathic pain [147]. This result suggests that neuropathic pain can alter the modulatory effect of adenosine on excitatory transmission in lamina II neurons.

In young hamster spinal lamina II neurons, it has been demonstrated that adenosine inhibits glutamatergic excitatory transmission by a pre- and postsynaptic mechanism; the latter action is due to the activation of K^+^ channels [148].

### 4.4. ATP Action

ATP activates P2 receptors classified into two subtypes, ionotropic P2X receptors and G protein-coupled metabotropic P2Y receptors; seven P2X receptors (P2X_1_-P2X_7_) and eight P2Y receptors (P2Y_1_, P2Y_2_, P2Y_4_, P2Y_6_, P2Y_11_, P2Y_12_, P2Y_13_ and P2Y_14_) have been characterized [149,150]. ATP in the lamina II would originate from primary-afferent central terminals or intrinsic neurons and/or glia cells in the spinal dorsal horn, generally being released from the cytoplasm to extracellular space through a nucleoside transporter.

In young hamster lamina II neurons, ATP (1-5 mM) induced a fast inward current (membrane depolarization) in a manner sensitive to a P2X- and P2Y-receptor antagonist, suramin (0.5 mM), and potentiated the peak amplitude of dorsal root-evoked EPSC [151]. A P2X- and P2Y-receptor antagonist, pyridoxal-phosphate-6-azophenyl-2',4'-disulfonic acid (PPADS; 10 μM), presynaptically reduced the peak amplitude of dorsal root-evoked EPSCs in young rat SDH neurons [152], although PPADS at 50 μM did not affect spontaneous and electrically-evoked EPSCs [153]. With respect to inhibitory transmission, ATP (10 μM) enhanced the release of glycine from nerve terminals in dissociated rat lamina II neurons attached with synaptic buttons in a manner sensitive to PPADS (10 μM) while resistant to *N*-ethylmaleimide (3 μM; a sulfhydryl alkylating agent having an ability to block G-protein functions), indicating an involvement of P2X receptors [154]. Intrathecal administration of PPADS did not produce any antinociceptive effects when examined by the tail-flick test [152]. These results demonstrate that ATP may play some role in modulating nociceptive transmission in the SDH, albeit this being unclear.

### 4.5. Noradrenaline Action

Adrenoceptors, which are activated by noradrenaline, are classified into at least three subtypes of α_1_ (1A-, 1B-, 1D-types), α_2_ (2A-, 2B-, 2C-types) and β (1-, 2-, 3-types), all of which are G protein-coupled metabotropic receptors. The α_1_ adrenoceptors activate an IP_3_/DAG system through PLC (by α subunit of G_q/11_ protein). On the other hand, the α_2_ adrenoceptors either inhibit adenylate cyclase (by α subunit of G_i/o_ protein) or regulate the activation of K^+^ or Ca^2+^ channels through βγ subunit of G_i/o_ protein, leading to their opening and closing, respectively. All of the β adrenoceptors activate adenylate cyclase (by α subunit of G_s_ protein; for review, see [155]). The α_1A/D_ and α_1B_ receptor mRNAs are present in the rat spinal cord, albeit being at a low density [156]. The α_2A_ and α_2C_ adrenoceptors are reported to exist in primary-afferent C-fiber central terminals and interneurons, respectively, in the SDH [157]. The rat SDH does not express β_1_ and β_2_ mRNAs [158]. There is a descending noradrenaline-containing fiber pathway from cell groups designated A5, A6 (nucleus locus ceruleus) and A7 (subceruleus) in the pons to the spinal dorsal horn [159,160]. Electrical stimulation of this pons region results in behavioral analgesia [161]. This noradrenaline pathway is also activated by systemically-administrated opioids [162] or electrical stimulation of the midbrain periaqueductal gray region [163]. Intrathecal administration of noradrenaline itself is known to have an antinociceptive effect when assessed by the tail-flick and hot-plate tests [164,165].

Noradrenaline acts on both pre- and postsynaptic sites in the spinal dorsal horn. In adult rat lamina II neurons, noradrenaline (10 μM) induced an outward current at –70 mV (membrane hyperpolarization); this effect was mimicked by an α_2_ agonist, clonidine (10 μM), and was blocked by an α_2_ antagonist, yohimbine (0.5 μM), indicating that this current response is due to the activation of α_2_ adrenoceptors [166,167]. Noradrenaline (10-100 μM) also enhanced the release of GABA and glycine to lamina II neurons from inhibitory neurons; this action was mimicked by an α_1_ agonist, phenylephrine (10-60 μM), and was inhibited by an α_1_ antagonist, prazocin (0.5 μM), and an α_1A_ antagonist, 2-(2,6-dimethoxyphenoxyethyl)aminomethyl-1,4-benzodioxane (WB-4101; 0.5 μM), indicating an involvement of α_1_, possibly α_1A_ adrenoceptors [166,168,169]. Noradrenaline (50 μM) did not affect sEPSC frequency and amplitude, but inhibited the release of electrically-evoked L-glutamate to lamina II neurons from primary-afferent Aδ-fiber and C-fiber central terminals, where Aδ-fiber EPSCs (50% peak amplitude reduction) were more sensitive to noradrenaline than C-fiber ones (28%) [170]. These inhibitory actions were mimicked by clonidine (10 μM) and an α_2A_ agonist, oxymetazoline (10 μM), and was blocked by yohimbine (1 μM), indicating an involvement of α_2_, possibly α_2A_ adrenoceptors [170]. A similar inhibitory action of clonidine has been reported for excitatory transmission in neurons existing in the outer layer of the lamina II [171]. All of the noradrenaline actions result in a decrease in the excitability of lamina II neurons. A β-adrenoceptor agonist, isoproterenol (40 μM), did not affect the inhibitory and excitatory transmission [166,170].

### 4.6. Serotonin Action

Serotonin (5-hydroxytryptamine, 5-HT) receptors are G protein-coupled metabotropic receptors except for the 5-HT_3_ receptor which is a cation-permeable channel. The metabotropic 5-HT receptors are composed of at least 5-HT_1_ (A-, B-, D-, E-, F-types), 5-HT_2_ (A-, B-, C-types), 5-HT_4_, 5-HT_5_ (A-, B-types), 5-HT_6_ and 5-HT_7_ receptors. The 5-HT_1_ and 5-HT_5_ receptors are negatively coupled to adenylate cyclase (by α subunit of G_i/o_ protein) while the 5-HT_4_, 5-HT_6_ and 5-HT_7_ receptors activate adenylate cyclase (by α subunit of G_s_ protein). The 5-HT_2_ receptor activates an IP_3_/DAG system through PLC (by α subunit of G_q/11_ protein; for review, see [172]). The 5-HT receptors are expressed within the spinal cord [173,174]; particularly, binding sites for (+)-hydroxy-2-(di-*n*-propylamino)tetralin (8-OH-DPAT, an agonist specific to the 5-HT_1A_ and 5-HT_7_ receptors) are expressed in the SDH including lamina II [174]. There are descending inhibitory serotonergic systems from the medullary raphe nuclei in the brainstem to the spinal dorsal horn [175,176]. Electrical stimulation of the nucleus raphe magnus releases 5-HT in the spinal dorsal horn [177]. Intrathecal application of 5-HT and 8-OH-DPAT resulted in antinociception when estimated using the tail-flick test [178].

In adult rat lamina II neurons, 5-HT (40 μM) induced either outward or inward currents at –70 mV, indicating a heterogeneity in 5-HT receptor subtype expression among different lamina II neurons [75,179]. The former (membrane hyperpolarization) was mimicked by 8-OH-DPAT (10 μM), and was completely blocked by a selective 5-HT_1A_ receptor antagonist, WAY 100635 (10 μM), indicating an involvement of 5-HT_1A_ receptors [180]. On the other hand, the latter (membrane depolarization) was observed in a small number of the neurons tested, and was mimicked by a 5-HT_3_ receptor agonist, 1-(*m*-chlorophenyl)-biguanide (mCPBG; 30 μM), indicating an involvement of 5-HT_3_ receptors [180]. When examined in morphologically-identified neurons, the 5-HT-induced outward currents were produced in vertical (21/34), small islet (11/34) and radial cells (2/34) while 5-HT-induced inward currents in islet (1/5) and small islet cells (4/5), indicating that 5-HT produces a membrane hyperpolarization in excitatory neurons, because most vertical cells are glutamatergic [180]. 5-HT (40 μM) also inhibited the release of L-glutamate to lamina II neurons from primary-afferent Aδ-fiber and C-fiber central terminals as noradrenaline did; inhibitions of Aδ-fiber and C-fiber responses were comparable in extent (each 39%). The action of 5-HT on C-fiber responses was mimicked by 8-OH-DPAT, but was not blocked by WAY 100635, indicating that 5-HT receptors existing in primary-afferent C-fiber central terminals appear to be a 5-HT_1A_-like type that is neither the 5-HT_1A_- nor the 5-HT_7_-type [9]. With respect to inhibitory transmission, 5-HT (100 μM) and mCPBG (30 μM) increased mIPSC frequency and amplitude in lamina II neurons; its amplitude increase was sensitive to TTX (1 μM). This result indicates that the inhibitory transmission enhancement is mediated by action potential production occurring as a result of 5-HT_3_ receptor-mediated depolarization [180]. Fukushima et al. [181] have reported spontaneous GABAergic transmission enhancement mediated by 5-HT_3_ receptors in adult mouse spinal SDH neurons. Moreover, 5-HT_3_ receptor activation in the spinal cord is shown to increase the release of GABA by using the microdialysis method [182].

### 4.7. Dopamine Action

Dopamine receptors, which are activated by dopamine, are classified into two subtypes of D1-like (D1 and closely-related D5) and D2-like (D2, closely-related D3 and D4) receptors, all of which are G protein-coupled metabotropic receptors. The D1 and D5 receptors activate adenylate cyclase (by α subunit of G_s_ protein) while the D2, D3 and D4 receptors are negatively coupled to adenylate cyclase (by α subunit of G_i/o_ protein; [183]). Both D1-like and D2-like receptors exist in the rat spinal cord at a high density [184]. There are descending dopaminergic pathways to the spinal dorsal horn from the periventricular posterior region (A11) of the hypothalamus in rats [185,186,187,188]. Intrathecally-administrated apomorphine (a D2 receptor agonist) inhibited thermally and chemically evoked noxious responses in rats ([189]; see [190] for review).

In adult rat lamina II neurons, dopamine concentration-dependently produced an outward current at –70 mV (membrane hyperpolarization; EC_50_ = 77.8 μM) in a manner resistant to TTX and CNQX and sensitive to intracellular GDP-β-S and extracellular Ba^2+^. The outward current produced by dopamine was mimicked by a D2-like receptor agonist quinpirole (30 μM) and depressed in amplitude by a D2-like receptor antagonist sulpiride (30 μM), indicating K^+^ channel opening (possibly by βγ subunit of G_i/o_ subunit) through D2-like receptor activation [191,192]. On the other hand, dopamine (100 μM) did not affect the frequency and amplitude of mEPSCs recorded in the presence of TTX [191].

### 4.8. Somatostatin Action

Somatostatin (a 14-amino acid peptide) receptors (SSTRs) are classified into at least five subtypes named SSTR_1_-SSTR_5_, all of which are coupled to G protein, resulting in inhibiting adenylate cyclase (by α subunit), opening K^+^ channels or closing Ca^2+^ channels (by βγ subunit) through G_i/o_ protein activation (see [193] for review). The SDH including lamina II contains SSTR-like immunoreactivity for the SSTR_1_-SSTR_3_ [194]. Somatostatin is expressed in rat DRG neurons [195] and somatostatin-positive fibers are localized at the highest density in the rat lamina II [196]. Intrathecal administration of somatostatin produced antinociceptive responses to noxious heat in cats [197] and inhibited nociceptive responses to subcutaneous formalin in rats [198].

In adult rat lamina II neurons, somatostatin produced an outward current at –60 mV (membrane hyperpolarization) in a manner resistant to TTX and in a concentration-dependent manner with an EC_50_ value of 0.82 μM [199,200]. Such a response to somatostatin (1 μM) was seen in about 50% of the neurons tested, indicating a heterogeneity in somatostatin receptor expression among different lamina II neurons [75]. The somatostatin activity was sensitive to intracellular Cs^+^ and TEA, and extracellular Ba^2+^, indicating an involvement of K^+^ channels. Consistent with this idea, somatostatin current exhibited an inwardly-rectifying property and reversed at a potential close to the equilibrium potential for K^+^. With respect to synaptic transmission, somatostatin (1 μM) had no effect on mEPSC frequency and amplitude, monosynaptic dorsal root-evoked EPSC amplitudes, and the frequency and amplitude of GABAergic and glycinergic mIPSCs [199]. A similar outward current produced by somatostatin has been reported in juvenile rat lamina II neurons [201]. It has not been examined what kinds of SSTRs are involved in the somatostatin-induced outward currents.

### 4.9. Cannabinoid Action

Cannabinoid receptors, which are activated by cannabinoids having an ability to alleviate pain, are classified into two subtypes, G_i/o_ protein coupled CB1 and CB2 receptors [202,203]. Intrathecal administration of a prototypical cannabinoid D^9^-tetrahydrocannabinol resulted in antinociception in the tail-flick test in adult rats [204]. A similar antinociceptive effect was produced by a mixed CB1/CB2 receptor agonist WIN55,212-2 (WIN-2) [205]. Such behavioral results are possibly mediated by CB1 receptors in the spinal dorsal horn, because localization of CB1 receptors has been demonstrated there by in situ hybridization [206], agonist binding [207] and immunohistochemistry [208]. A selective CB1-receptor antagonist SR141716A facilitated nociceptive responses of rat spinal dorsal horn neurons [209].

In adult rat lamina II neurons, an endocannabinoid *N*-arachidonoylethanolamide (anandamide; 0.01-10 μM) and WIN-2 (5-10 μM) had no effect on holding currents at –70 mV, sEPSC frequency and amplitude while reducing monosynaptically-evoked Aδ-fiber and C-fiber EPSC amplitudes; the former reduction (32% reduction by 10 μM anadamide) was larger than the latter (17% reduction) [210]. Although anandamide is thought to be an endogenous ligand of TRPV1 channels in vascular preparations [211], the anadamide activity in lamina II neurons was due to the activation of cannabinoid receptors but not TRPV1 channels, because the actions of anandamide on excitatory transmission were quite different from those of capsaicin [60,61]. A similar inhibitory action of anandamide on excitatory transmission has been shown in juvenile rat lamina II neurons [212]. With respect to inhibitory transmission, anandamide (10 μM) reduced focally-evoked GABAergic and glycinergic IPSC amplitudes in a manner sensitive to SR141716A (5 μM) in adult rat lamina II neurons [213]. Since anandamide (10 μM) reduced the frequency of GABAergic and glycincergic sIPSC without a change in the amplitude, its activities on evoked inhibitory transmission were pre- but not post-synaptic in origin [213]. Similar actions on inhibitory transmission were induced by WIN-2 (5 μM) and an endogenous cannabinoid-receptor agonist 2-arachydonoyl glycerol (20 μM; [214]) [213].

### 4.10. Galanin Action

Galanin (a 29/30-amino acid peptide), which was first extracted from porcine upper intestines [215], was reported to extensively exist in the peripheral and central nervous systems (see [216] for review). Galanin serves as a neurotransmitter or neuromodulator in various physiological functions such as feeding and pain (see [216,217,218] for review). There are three subtypes of G protein-coupled metabotropic receptor (GalR1, 2, 3: coupled to G_i/o_; G_i/o_ or G_q/11_; and G_i/o_ protein; respectively) for galanin [216]. There is much evidence for the idea that galanin plays a role in regulating nociceptive transmission to the spinal dorsal horn from the periphery. First, galanin immunoreactivity, GalR1, 2, 3 mRNAs and proteins are located in the rat DRG and the spinal dorsal horn [219,220,221,222,223,224,225]. The expression of galanin was upregulated in DRG neurons after nerve injury and in dorsal horn neurons after inflammation [216,218]. For example, peripheral inflammation induced by the injection of carrageenan into the hindpaw of rats increased the number of galanin mRNA-positive neurons in the spinal SDH [226]. Second, intrathecally administrated galanin modulated nociceptive responses in rats [227,228,229,230]. Transgenic mice overexpressing galanin in a population of DRG neurons exhibited nociceptive responses different from those of wild-type controls [231]. The intrathecal administration of galanin produced such a biphasic effect, as nociception at low doses and antinociception at high doses [229,230].

In adult rat lamina II neurons, galanin (0.03 μM) increased the frequency of sEPSC without a change in its amplitude, indicating a presynaptic effect; this action was concentration-dependent with an EC_50_ value of 2.0 nM. This effect reduced in extent in Ca^2+^-free or a voltage-gated Ca^2+^-channel blocker La^3+^ (30 μM)-containing Krebs solution, and was mimicked by a GalR2/R3 agonist galanin 2-11 [229] but not a GalR1 agonist M617 (galanin(1-13)-Gln^14^-bradykinin (3-9)amide [232]; each 0.03 μM). Galanin also produced in a concentration-dependent way an outward current at –70 mV (membrane hyperpolarization) with an EC_50_ value of 44 nM, a value larger than that for sEPSC frequency increase. This outward current was mimicked by M617 but not galanin 2-11 (each 0.1 μM). Moreover, galanin (0.1 μM) reduced monosynaptically-evoked Aδ-fiber and C-fiber EPSC amplitudes; the former reduction (35%) was larger than the latter (12%). A similar action was produced by galanin 2-11 but not M617 (each 0.1 μM). Spontaneous and focally-evoked inhibitory transmissions mediated by GABA and glycine were unaffected by galanin (0.1 μM). These results indicate that galanin at lower concentrations enhances the spontaneous release of L-glutamate from nerve terminals due to increased intracellular Ca^2+^ concentration by Ca^2+^ entry from external solution following GalR2/R3 activation while galanin at higher concentrations also produces a membrane hyperpolarization by activating GalR1 receptors. Moreover, galanin reduces L-glutamate release onto lamina II neurons from primary-afferent central terminals by activating GalR2/R3 receptors [233]. These effects could contribute to at least a part of the biphasic behavioral effect of galanin. Alier et al. [234] have reported an inhibition by galanin of excitatory transmission evoked in lamina II neurons by stimulating the dorsal root entry zone in young adult rats.

### 4.11. Substance P Action

Substance P (a 11-amino acid peptide) is a member of tachykinin family together with neurokinins A and B. Neurokinin (NK) receptors, which are activated by tachykinins, are classified into three subtypes of NK-1, NK-2 and NK-3, all of which are G-protein coupled metabotropic receptors, leading to the activation of PLC (by α subunit of G_q/11_ protein). Agonists most sensitive to the NK-1, -2 and -3 receptors are substance P, neurokinin A and neurokinin B (all of which are small peptides sharing a common amino acid sequence at their carboxy terminal), respectively (for review, see [235]). It is well-known that the activation of the NK1 receptor by substance P depolarizes membranes of dorsal horn neurons, leading to nociception [236] and that further such depolarization serves as a relief of NMDA receptors from a voltage-dependent block by Mg^2+^ (see [11]) which in turn results in sensitization called wind-up, a progressive increase in the number of action potentials evoked per stimulus in response to a repetitive stimulation of primary-afferent C-fibers [237,238,239].

Although the lamina II has the highest density of substance P-containing primary-afferent C-fiber terminals [240], this peptide seemed not to be involved in synaptic modulation in adult rat lamina II neurons, because substance P (1 μM) did not induce any responses [241] and a repeated stimulation (20 Hz for 1 sec) of primary-afferent C-fibers which was expected to release neuropeptides such as substance P did not produce any slow synaptic responses [26]. This result is consistent with the observation that there are few NK1 receptor-like immunoreactive neurons in the lamina II [242]. It may be possible that substance P released from the C-fiber endings acts on dendrites of neurons whose cell bodies exist in IV/V laminae deeper than the lamina II, resulting in membrane depolarization in deep dorsal horn neurons, as reported previously [243]. Projection neurons in the lamina I have a depolarizing response to substance P that plays a role in the induction of synaptic plasticity occurring there [244].

### 4.12. Bradykinin Action

Bradykinin (a 9-amino acid peptide) is locally produced from kininogens by kallikreins and kininases at the site of the injured and inflamed tissue. Bradykin receptors are classified into two subtypes of injury-induced B_1_ receptor and constitutively-expressed B_2_ receptor, both of which are G-protein coupled metabotropic receptors, leading to the activation of PLC (by α subunit of G_q/11_ protein; see [245] for review). Although the B_1_ and B_2_ receptors are located in the peripheral terminals of primary-afferent neurons and involved in peripheral sensitization (reduction in threshold for receiving nociceptive stimuli) ([246]; see [247] for review), there is evidence showing that bradykinin plays a role in modulating nociceptive transmission in the spinal dorsal horn. Intrathecal administration of a B_2_-receptor antagonist reduced the second phase of nociceptive responses to intraplantar formalin in rats [248] and intrathecally-administrated B_1_- and B_2_-receptor agonists produced thermal hyperalgesia measured by the hot-plate test in mice [249].

In adult rat lamina II neurons, bradykinin but not a B_2_-receptor agonist des-Arg^9^-bradykinin (each 10 μM) increased both bath-applied AMPA and NMDA responses that were measured at –70 and –40 mV (at the latter potential, a relief of blockade by Mg^2+^ of NMDA receptor-channels is expected; see [11]), respectively; the increases in the AMPA and NMDA responses were reduced in extent by intracellular GDP-β-S. These results indicate an increase in a sensitivity of AMPA and NMDA receptors to L-glutamate, caused by B_1_ receptor activation by bradykinin. Consistent with this postsynaptic effect, bradykinin (10 μM) increased the peak amplitude of monosynaptically-evoked primary-afferent Aδ-fiber and C-fiber EPSCs. Moreover, bradykinin but not des-Arg^9^-bradykinin (each 10 μM) increased the frequency and amplitude of mEPSC recorded in the presence of TTX, indicating not only postsynaptic but also presynaptic facilitatory actions of bradykinin, mediated by B_1_ receptors. This action was not accompanied by a change in holding currents at –70 mV. Such a glutamatergic transmission enhancement could explain the observation that intrathecal administration of bradykinin produced thermal hypersensitivity that was suppressed by an NMDA-receptor antagonist MK-801 [250].

### 4.13. Neuropeptide Y Action

Neuropeptide Y (a 36-amino acid peptide; C-terminal amidated one), which extensively exists in the peripheral and central nervous systems [251], plays a role in various physiological functions such as feeding and pain ([252,253]; see [254] for review). Neuropeptide Y receptors, which are activated by neuropeptide Y, are classified into at least six subtypes of Y1-Y6, all of which are G protein (G_i/o_ protein)-coupled metabotropic receptors [255]. There is much evidence for the idea that neuropeptide Y plays a role in regulating nociceptive transmission to the spinal dorsal horn from the periphery. First, neuropeptide Y-like immunoreactivity is densely located in the laminae I-II [256], Y1 receptors are expressed in small-sized DRG neurons and the spinal dorsal horn [257], and Y2 receptors are located in large-sized DRG neurons in rats [258]. Second, intrathecally-administrated neuropeptide Y produced antinociception in the hot-plate test in rats [259]. Third, Y1 receptor-knockout mice exhibited hyperalgesia in response to acute thermal, cutaneous and visceral chemical stimuli, while completely lacking the analgesic effects of neuropeptide Y [260].

In 33% of the adult rat lamina II neurons examined, neuropeptide Y (1 μM) produced an outward current at –60 mV (membrane hyperpolarization) in a manner resistant to TTX. Such a paucity of responsive neurons also shows a heterogeneity in neuropeptide Y receptor expression among different lamina II neurons [75]. The neuropeptide Y activity was sensitive to intracellular Cs^+^ and TEA, and extracellular Ba^2+^, indicating an involvement of K^+^ channels. Consistent with this idea, neuropeptide Y current exhibited an inwardly-rectifying property and reversed at a potential close to the equilibrium potential for K^+^. Moreover, neuropeptide Y activity was sensitive to intracellular GDP-β-S, was mimicked by a Y1-receptor agonist [Leu^31^,Pro^34^]-neuropeptide Y (1 μM) and suppressed by a Y1-receptor antagonist BIBP 3226 (1 μM), indicating Y1 receptor activation. With respect to synaptic transmission, neuropeptide Y (1 μM) had no effect on sEPSC frequency and amplitude, monosynaptic dorsal root-evoked EPSC amplitudes, GABAergic and glycinergic mIPSC frequency and amplitude, and focally-evoked GABAergic and glycinergic IPSC amplitudes [261].

### 4.14. Phospholipase A_2_ Activation Action

Phospholipase A_2_ (PLA_2_) is thought to play a pivotal role in a variety of physiological functions including nociception through the production of arachidonic acid which is one of fatty acids released from the *sn*-2 position of membrane phospholipids by PLA_2_ activation (for review, see [262]). The arachidonic acid is involved in regulating neuronal functions as a result of the synthesis of eicosanoids such as prostanoids (for review, see [263]) or without conversion to metabolites (for review, see [264]). The spinal cord contains the small molecular-weight secreted PLA_2_ (sPLA_2_) and the large molecular-weight cytosolic PLA_2_ (for review, see [265] and [266]).

Melittin (a 26-amino acid basic peptide; a major component of the bee venom [267]) is known to be an in vitro activator of sPLA_2_ with no effect on cytosolic PLA_2_ [268,269,270]. In adult rat lamina II neurons, melittin (1 μM) did not change holding current at –70 mV while increasing the frequency and amplitude of sEPSC. This frequency increase was concentration-dependent with an EC_50_ value of 1.1 μM. Melittin activity disappeared in the presence of a selective PLA_2_ inhibitor 4-bromophenacyl bromide (4-BPB, 10 μM; [271]) while being unaffected by TTX (0.5 μM), a cyclooxygenase inhibitor indomethacin (100 μM) and a lipoxygenase inhibitor nordihydroguaiaretic acid (NDGA; 100 μM), indicating an involvement of sPLA_2_ activation and possibly arachidonic acid but not its metabolites produced by cyclooxygenase and lipoxygenase [272]. Melittin (1 μM) also increased the frequency and amplitude of sIPSC; the effect of melittin on GABAergic but not glycinergic sIPSC was not seen in the presence of TTX (0.5 μM) and CNQX (10 μM). EC_50_ values for melittin to increase glycinergic sIPSC frequency and amplitude were 0.73 and 0.64 μM, respectively [28]. 4-BPB, another PLA_2_ inhibitor aristolochic acid (100 μM; [273]), and NDGA (100 μM) but not indomethacin (100 μM) inhibited the facilitatory action of melittin on glycinergic transmission. These results indicate that the GABAergic transmission enhancement is mediated by glutamate-receptor activation and neuronal activity increase, possibly owing to excitatory transmission enhancement, while the glycinergic one is due to PLA_2_ and subsequent lipoxygenase activation [28]. The GABAergic transmission enhancement was attributed to actions of acetylcholine (ACh) and noradrenaline which activate ACh receptors (AChRs; nicotinic and muscarinic types; see below) and α_1_ adrenoceptors, respectively ([169]; see [274] for review). With respect to arachidonic acid’s metabolites, prostaglandin E_2_ (10 μM) is reported to produce a membrane depolarization with no effect on excitatory transmission in a minority of the adult rat lamina II neurons examined [275]. In mouse spinal SDH neurons, prostaglandin E_2_ increased mEPSC frequency [276]. There appeared to be a difference in prostaglandin E_2_ activity between animals used.

### 4.15. Acetylcholine Action

AChRs are classified into ionotropic nicotinic AChRs (nAChRs) and metabotropic muscarinic AChRs (mAChRs). The nAChRs in the central nervous system are composed of either a combination of α (2-6 types) and β (2-4 types) subunits or a homomer of α7-α9 subunits (for review, see [277]), while the mAChRs are composed of M1, M2, M3, M4 and M5 receptors. The M1, M3 and M5 receptors activate an IP_3_/DAG system through PLC (by G_q/11_ protein), while the M2 and M4 receptors inhibit adenylate cyclase (by G_i/o_ protein; [278]). The adult rat lamina II contains α3, α4, α5 and β2 subunit mRNAs [279,280]. A high density of mAChRs which bind [^3^H]-pirenzepine has been demonstrated in the lamina II by autoradiographic studies in adult rats [281]. Choline acetyltransferase (an enzyme which synthesizes ACh)-immunoreactive neurons in the lamina III are frequently presynaptic to lamina II neurons [282] and this enzyme co-localizes with GABA in the spinal dorsal horn [283], suggesting that ACh in the lamina II originates from inhibitory neurons in the spinal dorsal horn. Intrathecal administration of a mAChR agonist carbamoylcholine or ACh esterase inhibitors (reversible ones: neostigmine and physostigmine; irreversible one: echothiophate) produced analgesia to noxious thermal stimuli in rats [284,285]. On the other hand, intrathecal application of nAChR agonists, nicotine, cytisine, A-85380 and epibatidine, resulted in nociception but the latter two drugs also produced antinociception in a manner sensitive to a broad-spectrum nAChR antagonist, mecamylamine, when evaluated by the paw-withdrawal test to heat, suggesting the presence of a type of nAChRs involved in antinociception in the spinal dorsal horn [286,287].

In adult rat lamina II neurons, (-)-nicotine (100 μM) or carbamoylcholine (10 μM) elicited an inward current at –70 mV (membrane depolarization) [20,288,289]. ACh also has an ability to enhance the release of GABA and glycine from the terminals of GABAergic and glycinergic neurons by the activation of nAChRs and mAChRs. (-)-Nicotine (100 μM) and (-)-cytisine (20 μM) enhanced the frequency and amplitude of GABAergic and glycinergic sIPSCs in a manner sensitive to mecamylamine (5 μM; [169,289]). Baba et al. [288] have demonstrated that carbamoylcholine (10 μM) and neostigmine (10 μM; which is expected to increase the concentration of ACh in the synaptic cleft) increase the frequency and amplitude of GABAergic sIPSC (also see [169]). These results suggest that antinociception produced by ACh is mediated by inhibitory neurotransmitters such as GABA and glycine released by the activation of nAChRs and mAChRs.

## 5. Similarity and Difference among Endogenous Neuromodulators in Antinociceptive Mechanisms at Cellular Levels in the Spinal Lamina II

Table 1 demonstrates a comparison of synaptic modulation produced by oxytocin, orexins A and B with those of other endogenous pain neuromodulators in rodent lamina II neurons. Membrane depolarization produced by oxytocin [20], and membrane depolarization and/or enhanced spontaneous L-glutamate release produced by orexins A and B [21,22], in lamina II neurons could increase the membrane excitability of these neurons. This depolarizing effect was distinct from those of analgesic neuropeptides (endomorphins [104], nociceptin [125,128], somatostatin [199,200], galanin [233] and neuropeptide Y [261]), adenosine [143], noradrenaline [167], 5-HT [9,180] and dopamine [191,192], which produced membrane hyperpolarization in lamina II neurons, leading to antinociception. Although oxytocin, orexins A and B produced no change or increase in L-glutamate release from nerve terminals in the lamina II [20,21,22], the spontaneous or electrically evoked release of L-glutamate from nerve terminals was inhibited by endomorphins [104,106], nociceptin [126], adenosine [142,144], noradrenaline (evoked release; [170]), 5-HT [9], cannabinoids (evoked release; [210]) and galanin (evoked release; [233]), resulting in antinociception. The idea that reduced primary-afferent central terminal L-glutamate release to lamina II neurons is involved in antinociception is supported by the observation that sEPSC frequency is increased by the activation of TRPV1 channels whose inhibitors produce antinociception when intrathecally administrated (for review, see [62,290,291]). Bradykinin, intrathecal administration of which produced nociception, increased L-glutamate release from nerve terminals and a sensitivity to L-glutamate of AMPA and NMDA receptors, both of which enhanced glutamatergic transmission in lamina II neurons [250]. Such a sensitivity increase of AMPA and NMDA receptors in lamina II neurons was produced by interleukin-1β [292] that produced a measure of nociception, i.e., wind-up (see above), when administrated intrathecally [293].

As with a GABA_B_-receptor agonist baclofen [27,241,294,295] and a metabolite of opioid tramadol, *O*-desmethyltramadol [296,297,298], clinically-used analgesics that produce antinociception by intrathecal administration generally hyperpolarize membranes and inhibit the release of L-glutamate from nerve terminals in the lamina II.

Interestingly, oxytocin, orexins A and B enhanced spontaneous inhibitory transmission [20,21,22]. As a facilitation of this transmission decreases the membrane excitability of lamina II neurons, this enhancement could account for the antinociceptive effect of oxytocin [43,44], orexins A and B [72,73,74], as suggested for antinociception produced by noradrenaline [166,168,169] and 5-HT [180,181]. Since noradrenaline and 5-HT also hyperpolarize membranes and reduce L-glutamate release from nerve terminals in the lamina II [9,167,170,180], these actions also could contribute to their antinociceptive actions together with enhanced inhibitory transmission. Consistent with the idea that GABAergic transmission enhancement is involved in antinociception produced by oxytocin, bicuculline suppressed antinociceptive responses produced by the electrical stimulation of hypothalamic paraventricular nucleus or oxytocin application [299].

On the other hand, nAChR and mAChR activation produced by ACh leads to a membrane depolarization, which in turn facilitates spontaneous inhibitory transmission in lamina II neurons, resulting in antinociception [169,288,289]. Thus, the antinociceptive mechanism of oxytocin appears to be similar to that of ACh. Lamina II neurons sensitive to oxytocin exhibited a membrane depolarization in response to (-)-nicotine and carbamoylcholine [20]. As a portion of lamina II neurons is sensitive to both oxytocin and orexin B [22], these hypothalamic neuropeptides and ACh may produce antinociception in a synergistic manner.

With respect to spontaneous inhibitory transmission, oxytocin and orexin A enhanced both GABAergic and glycinergic transmission [20,21], while orexin B enhanced glycinergic but not GABAergic transmission in the majority of lamina II neurons examined [22]. In general, both GABAergic and glycinergic transmission in lamina II neurons are modulated by many pain modulators in a similar manner (no change with DAMGO [101], nociceptin [126], somatostatin [199], galanin [233] and neuropeptide Y [261]; inhibition with adenosine [145] and anandamide [213]). On the other hand, GABAergic transmission was affected by noradrenaline in a distinct manner from that of glycinergic transmission [168] and PLA_2_ activation produced by melittin resulted in glycinergic but not GABAergic transmission enhancement in the presence of TTX [28,169]. 5-HT_3_ receptor activation in the spinal cord increased the release of GABA but not glycine, measured by the microdialysis method [182]. There are neurons exhibiting synaptically-evoked glycine but not GABA responses in the adult rat SDH [300]. It remains to be investigated why there is a difference in synaptic modulation by endogenous pain modulators between GABAergic and glycinergic transmission in lamina II neurons.

Since neuropathic or inflammatory pain can drastically change the level of endogenous neuromodulator or the function of receptor for neuromodulator, as seen in the cases of nociceptin (see [113] for review), adenosine and galanin (see Section 4.2, Section 4.3 and Section 4.10), such changes also will have to be taken into consideration in pain neuromodulators’ effects on nociceptive transmission.

Oxytocin, orexins A and B exclusively activate oxytocin, orexin-1 and orexin-2 receptors, respectively, while many of other antinociceptive modulators activate several receptor subtypes whose activation results in distinct synaptic modulation. Therefore, pain therapy by oxytocin or orexins in the spinal cord level may have an advantage over other antinociceptive modulators in activating only one kind of receptor leading to inhibitory transmission enhancement.

## 6. Conclusions

The present review article demonstrated that the antinociceptive effects of hypothalamic neuropeptides (oxytocin, orexins A and B) are produced by membrane depolarization and/or increased spontaneous release of L-glutamate from nerve terminals, both of which result in action potential production that leads to enhanced spontaneous inhibitory transmission by activating their specific receptors in the adult rat spinal lamina II. Such a mechanism was partly similar to those of the other analgesics in the lamina II. There will be a particular spinal dorsal horn circuitry that produces a net inhibition of the projection neurons transferring nociceptive information to the brain as a result of synaptic modulation produced by the hypothalamic neuropeptides in the lamina II. Such a circuitry remains to be revealed.

It is likely that not only the spinal lamina II but also other nervous systems are involved in the antinociceptive effects of oxytocin, orexins A and B. For instance, it is likely that oxytocin-containing hypothalamic neurons project to neurons of deep layers of the spinal cord, leading to antinociception (see [301]). This idea may possibly also apply to the case of orexins, because an abundant distribution of orexins is present in both superficial and deep layers in the lumbar segment of the rat spinal cord [69]. Alternatively, orexin A possibly acts on brain stem neurons including locus coeruleus or periaqueductal gray neurons that activate the descending pain inhibitory pathway to the spinal dorsal horn, leading to antinociception [302,303]. In future, it would be necessary to examine the involvement of neural pathways other than the hypothalamus-spinal lamina II pathway in antinociception produced by oxytocin and orexins.

## Figures and Tables

**Table 1 pharmaceuticals-12-00136-t001:** Comparison of synaptic modulation produced by oxytocin, orexins A and B with those of other endogenous pain neuromodulators in rodent spinal lamina II neurons.

Endogenous Neuromodulators	Resting Membrane Potential	Glutamatergic Excitatory Transmission	GABAergic Spontaneous Inhibitory Transmission	Glycinergic Spontaneous Inhibitory Transmission	References
Oxytocin ^*1^	Depolarization	No change	Facilitation (sensitive to TTX)	Facilitation (sensitive to TTX)	[20]
Orexin A ^*1^	Depolarization	Facilitation	Facilitation (sensitive to TTX)	Facilitation (sensitive to TTX)	[21]
Orexin B ^*1^	Depolarization	Facilitation	No change	Facilitation (sensitive to TTX)	[22]
Endomorphins ^*1^	Hyperpolarization	Depression	No change	No change	[104,106]
Nociceptin ^*1^	Hyperpolarization	Depression	No change	No change	[125,126]
Adenosine ^*1^	Hyperpolarization	Depression	Depression	Depression	[142,143,144,145]
ATP	Fast depolarization	Facilitation	−	Facilitation	[151,154]
Noradrenaline ^*1^	Hyperpolarization	No change (spontaneous) Depression (evoked)	Facilitation	Facilitation	[166,167,168,170]
Serotonin (5-HT) ^*1^	HyperpolarizationDepolarization	Depression	Facilitation	Facilitation	[9,180,181]
Dopamine ^*1^	Hyperpolarization	No change	−	−	[191,192]
Somatostatin ^*1^	Hyperpolarization	No change	No change	No change	[199,200]
Cannabinoids ^*1^	No change	No change (spontaneous) Depression (evoked)	Depression	Depression	[210,213]
Galanin ^*2^	Hyperpolarization	Facilitation (spontaneous) Depression (evoked )	No change	No change	[233]
Substance P ^*3^	No change	No change	−	−	[241]
Bradykinin ^*3^	No change	Facilitation	−	−	[250]
Neuropeptide Y ^*1^	Hyperpolarization	No change	No change	No change	[261]
Phospholipase A_2_ activator	No change	Facilitation	Facilitation (sensitive to TTX)	Facilitation (resistant to TTX)	[28,169,272]
Acetylcholine (nicotinic) ^*1^	Depolarization	No change	Facilitation	Facilitation	[20,289]
Acetylcholine (muscarinic) ^*1^	Depolarization	No change	Facilitation	Facilitation	[28,288]

Here, when neurons responsive to neuromodulators exhibit several effects, the main effect of them is shown. ^*1^: neuromodulator that produces antinociception; ^*2^: production of both antinociception and pronociception in a manner dependent on its concentration; ^*3^: neuromodulator that produces pronociception (see the text for its detail). GABA: γ-aminobutyric acid; TTX: tetrodotoxin; ATP: adenosine 5’-triphosphate; −: data are not available, to my knowledge.

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
