# Peer review of "Cellular Mechanisms for Antinociception Produced by Oxytocin and Orexins in the Rat Spinal Lamina II—Comparison with Those of Other Endogenous Pain Modulators"

_pharmaceuticals, 2019, doi:10.3390/ph12030136_

Round 1

Reviewer 1 Report

This review provides a lot of information about a number of neuromodulators that act on dorsal horn nociceptive circuits, which is very useful information, but not what I expected when I read the abstract. The abstract is focused on oxytocin and orexins A and B and very briefly mentions that these will be compared to other endogenous analgesics, yet there is very little comparison. Both the title and abstract state that these modulators will be compared with other endogenous analgesics – some of the neuromodulators listed (eg: SP, bradykinin, PLA2) are pronociceptive.

Because there are so many topics within this one review, there are parts where the information isn’t very complete. I would have liked to see less topics with more detail, or more of a focus for each topic. For example, section 2, lines 45-47: plastic changes in excitatory inputs change in most pain models, not just intraplantar CFA and ovariectomy. Then in lines 48-53 only reductions of KCC2 and GAD are provided as explanations for disinhibition in the dorsal horn, with no mention of GlyRa3 (which is relevant to the CFA model), or other changes in GABAergic and glycinergic signalling that have recently been described. Lines 50-51: Should it say a decrease in expression of KCC2 in pain, rather than local blockade or knockdown? Lines 57-58: I would have thought that lamina II was chosen because of its importance in pain processing, rather than the ease of visually identifying the region.

I was confused about whether this was a review or original research paper when I got to section 2 as the author has included methods. Lines 72-77 describe the recording conditions – is this the same for all papers reviewed? This explanation doesn’t hold for experiments where neurons are chloride loaded. Lines 78-87 gives a recipe for a recording solution and again I wonder why this is included in a review. I found the suggestion that excitatory neurotransmission is mediated entirely through AMPA a bit misleading, as NMDA-R activity is really important in pain. You wouldn’t see them at -70mV due to Mg block, but they’re there. Lines 100-101: Have you taken into consideration the failure of C vs A fibres at this frequency? Lines 105-106: Spontaneous EPSCs that are TTX resistant are mEPSCs (they are not a product of AP activity in the presynaptic neuron). Lines 119-121: I don’t understand why the age and weight of rats is stated here, in a review paper.

Line 213- oxytocin is superfused for 3 m – should this be minutes rather than metres?

Many of the articles reviewed (other than self-cites) here are from a decade or three ago. Its important that the original research is cited, but there are places where the sections lack recent studies, for example lines 479-80: antinociception is also due to activation of the adenosine A3 receptor.

Another concern throughout this review is that we now know that lamina II neurons are very heterogeneous and often modulators that increase/decrease activity of some have no effect or the opposite on others. This needs to be considered in some of the sections, or noted at least.

Author Response

Thank you very much for reviewing my manuscript and providing valuable comments for this revision.  I would like to reply to your comments as follows:

This review provides a lot of information about a number of neuromodulators that act on dorsal horn nociceptive circuits, which is very useful information, but not what I expected when I read the abstract. The abstract is focused on oxytocin and orexins A and B and very briefly mentions that these will be compared to other endogenous analgesics, yet there is very little comparison. Both the title and abstract state that these modulators will be compared with other endogenous analgesics – some of the neuromodulators listed (eg: SP, bradykinin, PLA2) are pronociceptive.

Response:

Thank you very much for your valuable opinion.  The aim of this review article was to compare the effects of oxytocin, orexins A and B on synaptic transmission in lamina II neurons with those of other pain modulators, as given in Table 1, since lamina II neurons play a pivotal role in regulating nociceptive transmission, although a detail comparison in cellular mechanisms could not be performed because of few information about the mechanisms.

As pointed out by you, SP, bradykinin and PLA2 are generally involved in pronociceptive transmission.  In title and a part of the text in the original manuscript, “analgesics” was replaced by “pain modulators”, as given in the title of Table 1.  Please see title, lines 24-25, 66, 69, 897-898 and 906 in the revised manuscript.

Because there are so many topics within this one review, there are parts where the information isn’t very complete. I would have liked to see less topics with more detail, or more of a focus for each topic. For example, section 2, lines 45-47: plastic changes in excitatory inputs change in most pain models, not just intraplantar CFA and ovariectomy. Then in lines 48-53 only reductions of KCC2 and GAD are provided as explanations for disinhibition in the dorsal horn, with no mention of GlyRa3 (which is relevant to the CFA model), or other changes in GABAergic and glycinergic signalling that have recently been described.

Response:

Thank you very much for your valuable opinion.  As pointed out by you, a possible involvement of glycine receptors in inflammatory pain had not been mentioned in the original manuscript.  To describe this possibility, I mentioned the following sentence:

“There is a difference between wild-type and Glra3-/- mice in behavior of pain model produced by complete Freund’s adjuvant injection, indicating an involvement of glycine-receptor a3 subunit (see [11] for review) in inflammatory pain [12]”.  Please see lines 50-52 in the revised manuscript.

Moreover, in order to mention other changes in GABAergic and glycinergic signalling that were recently described, I gave the following sentence:

Recently, Medrano et al. [15] have suggested that a shift in the reversal potential for chloride is an important component of a loss of inhibitory tone in neuropathic pain mouse models produced by nerve injury”.  Please see lines 55-57 in the revised manuscript.

I have added to References the following two publications:

12. Harvey, V.L.; Caley, A.; Müller, U.C.; Harvey, R.J.; Dickenson, A.H. A selective role for a3 subunit glycine receptors in inflammatory pain. Front. Mol. Neurosci. 2009, 2, 14.

15. Medrano, M.C.; Dhanasobhon, D.; Yalcin, I.; Schlichter, R.; Cordero-Erausquin, M. Loss of inhibitory tone on spinal cord dorsal horn spontaneously and nonspontaneously active neurons in a mouse model of neuropathic pain. Pain 2016, 157, 1432-1442.

Lines 50-51: Should it say a decrease in expression of KCC2 in pain, rather than local blockade or knockdown?

Response:

Thank you very much for your valuable opinion.  According to your advice, “local blockade or knockdown” has been changed to “a decrease in expression”; please see line 52 in the revised manuscript.

Lines 57-58: I would have thought that lamina II was chosen because of its importance in pain processing, rather than the ease of visually identifying the region.

Response:

Thank you very much for your valuable opinion.  I have deleted “because” in the original manuscript.  Please see line 61 in the revised manuscript.

I was confused about whether this was a review or original research paper when I got to section 2 as the author has included methods. Lines 72-77 describe the recording conditions – is this the same for all papers reviewed? This explanation doesn’t hold for experiments where neurons are chloride loaded. Lines 78-87 gives a recipe for a recording solution and again I wonder why this is included in a review.

Response:

Thank you very much for your valuable opinion.  According to reviewer 2’s instruction, lines 72-87 in the original manuscript have been deleted.

I found the suggestion that excitatory neurotransmission is mediated entirely through AMPA a bit misleading, as NMDA-R activity is really important in pain. You wouldn’t see them at -70mV due to Mg block, but they’re there.

Response:

Thank you very much for your valuable opinion.  According to your advice, I have added a sentence, “NMDA receptor-channels are activated at more positive potentials than -70 mV (see Sections 4.11 and 4.12)”.  Please see lines 81-82 in the revised manuscript.

Lines 100-101: Have you taken into consideration the failure of C vs A fibres at this frequency?

Response:

Thank you very much for your comment.  Ad-fiber EPSCs were thought to be monosynaptic when the latency remains constant and there was no failure during stimulation at 20 Hz for 1 sec, while C-fiber EPSCs were thought to be monosynaptic when there was no failure during repetitive stimulation at 2 Hz for 10 sec.  Such an identification of Ad-fiber and C-fiber responses was based on data obtained from rat dorsal root ganglion neurons, as seen in Fig. 1 of Ataka et al. (2000)’s paper (ref. 25 in the revised manuscript).  I think that the sentence about this content is unnecessary to be revised.

Lines 105-106: Spontaneous EPSCs that are TTX resistant are mEPSCs (they are not a product of AP activity in the presynaptic neuron).

Response:

Thank you very much for your comment.  Spontaneous EPSCs recorded from lamina II neurons in our spinal cord slices are not affected in frequency and amplitude by TTX, as seen in Fig. 2B1 of Jiang et al. (2014)’s paper (ref. 20 in the revised manuscript) and in Fig. 3Cb of Wang et al. (2018)’s paper (ref. 22 in the revised manuscript).  On the other hand, spontaneous EPSCs recorded from lamina II neurons in vivo are inhibited in frequency and amplitude by TTX, as seen in Fig. 3D of Sonohata et al. (2004)’s paper (ref. 156 in the revised manuscript).  Thus, there appears to be a deafferentiation in the spinal cord slices used, as mentioned in line 107 in the original manuscript.  I think that the sentence about this content is unnecessary to be revised.

Lines 119-121: I don’t understand why the age and weight of rats is stated here, in a review paper.

Response:

Thank you very much for your valuable opinion.  According to reviewer 2’s instruction, lines 119-121 in the original manuscript have been deleted.

Line 213- oxytocin is superfused for 3 m – should this be minutes rather than metres?

Response:

Thank you very much for pointing out this mistake.  This has been corrected.  Please see line 200 in the revised manuscript.

Many of the articles reviewed (other than self-cites) here are from a decade or three ago. Its important that the original research is cited, but there are places where the sections lack recent studies, for example lines 479-80: antinociception is also due to activation of the adenosine A3 receptor.

Response:

Thank you very much for your valuable opinion.  As instructed by you, I mentioned an involvement of adenosine A3 receptors in antinociception, as follows:

“Recent studies have demonstrated an effectiveness of adenosine A3 receptor activation in alleviating neuropathic pain in rodents [130]; this antinociceptive effect appears to be due to an inhibition of enhanced microglial activation in the spinal dorsal horn [131].”  Please see lines 470-472 in the revised manuscript.

I have added to References the following two publications:

130. Chen, Z.; Janes, K.; Chen, C.; Doyle, T.; Bryant, L.; Tosh, D.K.; Jacobson, K.A.; Salvemini, D. Controlling murine and rat chronic pain through A3 adenosine receptor activation. FASEB J. 2012, 26,1855-1865.

131. Terayama, R.; Tabata, M.; Maruhama, K.; Iida, S. A3 adenosine receptor agonist attenuates neuropathic pain by suppressing activation of microglia and convergence of nociceptive inputs in the spinal dorsal horn. Exp. Brain Res. 2018, 236, 3203-3213.

Another concern throughout this review is that we now know that lamina II neurons are very heterogeneous and often modulators that increase/decrease activity of some have no effect or the opposite on others. This needs to be considered in some of the sections, or noted at least.

Response:

Thank you very much for your valuable opinion.  Although I mentioned a heterogeneity of lamina II in orexin B actions in the original manuscript (lines 262-263 and 316-317), this revised version pointed out a heterogeneity of lamina II neurons in the actions of endomorphins, serotonin, somatostatin and neuropeptide Y (lines 408-412, 567, 617-618 and 762-763, respectively, in the revised manuscript).

Reviewer 2 Report

The paper is supposed to be a review. However the author oscillates between original paper and review style. This oscillation introduces in the paper parts that are not necessary. Particularly following parts have nothing to do in a review and should be suppressed:

Line 72-87

Line 19-121

Author Response

The paper is supposed to be a review. However the author oscillates between original paper and review style. This oscillation introduces in the paper parts that are not necessary. Particularly following parts have nothing to do in a review and should be suppressed:

Line 72-87

Line 19-121

Reply to Comments:

Thank you very much for reviewing my manuscript and providing valuable comments for this revision.  I would like to reply to your comments as follows:

I have deleted lines 72-87 and lines 119-121, only a part of which information has been added in the text.

Reviewer 3 Report

The review by Kumamoto attempts to compare the electrophysiological actions of oxytocin and orexins within the spinal cord to various other neuromodulators that alter nociception. This is a huge undertaking and there is a lot of good information in this review however I feel the overall message is being let down by the lack of structure.

The anti-nociceptive properties of oxytocin and orexins A and B are paradoxical when their cellular actions are considered. They directly depolarise superficial spinal cord neurons and increase excitatory transmission, all of which would be expected to facilitate nociception. However they also differentially facilitate GABAergic and glycinergic transmission, which would be expected to reduce nociception (i.e. analgesic). Presumably the net anti-nociceptive properties of these neuropeptides are due to the relative distribution of their receptors and perhaps selective preferential coupling to effectors within particular spinal cord neurons/circuits that promote inhibition of nociceptive signalling. If this is the case then the complex circuitry and heterogeneity of spinal cord neurons has to be considered in relation to neuromodulator effect (i.e. does the neuromodulator target specific subsets of cells? Is facilitation of excitatory inputs within a particular spinal cord circuit that actually results in net inhibition of LI projection neurons?). In addition, neuropathic or inflammatory pain can drastically alter the levels of endogenous neuromodulator or receptor expression (e.g. see Imlach et al, 2015 Mol Pharmacol, 88 (3), 460-468, PMID 26104547 showing increased adenosine tone and enhanced A1R sensitivity at excitatory synapses). These are important points that govern the action of each neuromodulator but were either not discussed or not clearly defined in the current review and this needs to be addressed. In this regard, it would be helpful for the reader to have a cartoon of the relevant spinal cord circuitry and the various different cell-types and inputs within it to refer to. It would also be helpful if the effects of each neuromodulator were discussed in the context of this circuitry.

Each of the neuromodulators discussed act at a GPCRs however the author only intermittently indicated which particular G-protein was engaged (i.e. Gi/o, Gs, Gq) and which intracellular pathways were upregulated by each neuromodulator. This needs to be clarified for all neuromodulators discussed. In addition, no mention of bg subunit signalling was alluded to and this is a key component of the direct membrane potential changes resulting from GIRK activation or CaV inhibition following GPCR activation, particularly those coupled to Gi/o. In fact, the strength of receptor coupling to each effector can markedly alter the outcome of receptor activation and this can vary between CNS regions and between neuronal subtypes within a single region. These points need to be discussed.

There are a number of incorrect statements or omissions that need to be amended throughout the manuscript:

1)      Page 2 line 48: ‘naïve’ should be changed to ‘sham’

2)      Page 2 line 95 to page 3 line 107: The author indicates TTX has no effect on spontaneous currents recorded in the spinal cord and that sEPSC/sIPSCs are the same as mEPSCs/mIPSCs. This is simply not true. sEPSC/sIPSCs include action-potential driven events (intrinsic or network activity) whilst mEPSC/mIPSCs represent spontaneous release in the absence of AP generation and/or propagation. They are two very different measures and do not equate the same thing. Therefore, the author needs to specify whether sPSC or mPSCs are being referred to throughout the manuscript.

3)      Page 8, lines 385-388: reduced opioid binding sites following dorsal rhizotomy or capsaicin treatment does not indicate opioid receptors are localised at both pre- and post-synaptic sites, it just indicates these treatments alter opioid receptor levels. Their subcellular localisation can only be determined either with EM or functionally via the various electrophysiological recordings detailed throughout the paper.

4)      Page 8 line 406: the author states sEPSC frequency in diminished by DAMGO & DPDPE and this indicated mu and delta opioid receptors are in the terminals of glutamatergic interneurons. Since sEPSCs are a measure of all spontaneous excitatory transmission, it is impossible to determine whether this is originating from interneurons or external inputs such as the primary afferent neurons.

5)      Page 9, line 420-421: the postsynaptic effects of opioids needs expanding. Opioids have a complex and selective action on subpopulations of dorsal horn neurons. For example the mu and delta receptor agonists produce an outward hyperpolarising current in distinct subsets of neurons within LII neurons but in LI neurons both agonists can have an effect in the same neuron.

6)      No mention of kappa opioid receptor or deltorphin effect in the spinal cord.

7)      Page 9, Lines 418 & 449: the author states GABA and glycine transmission is not affected by DAMGO but please see Kerchner & Zhuo (2002), J Neurophysiology, 88(1), 520-522.

8)      Page 9, Line 432: “nociceptin peptide and mRNA” is likely a misprint. Since nociceptin is cleaved from a precursor protein (pre-pronociceptin), it does not have dedicated mRNA, rather pre-pronociceptin mRNA levels are used as an indicator of nociceptin transcription.

9)      Several detailed reviews have already been written on the electrophysiological actions of nociceptin and opioids (e.g. see PMIDs: 10998530, 30838458 & 24762001), these could be referenced to strengthen these sections.

10)   The statement “indicating heterogeneity of lamina II neurons” has been used frequently throughout the manuscript (e.g. P11, line 568; p12 line619…). This statement isn’t incorrect – LII neurons are diverse but it can be expanded upon. For example, P11, line 568: the varying response to 5HT (inward vs outward current) indicates heterogeneity in 5HT receptor subtype expression between different LII neurons and indeed Lu & Perl (see PMID 17463043) show an inward/outward current can vary between LII (& LI) neuronal subtypes (central, radial, vertical, islet). Being more specific on this heterogeneity where possible will also address my first comment.

11)   The table should include whether the neuromodulators are pro- or anti-nociceptive when then are delivered intrathecally, systemically or I.C.V (this can vary depending on the neuromodulator and delivery method).

12)   Page 18 line 905: “there are glycinergic but not GABAergic synapses in the adult rat SDH” suggests adult rats have no GABAergic transmission in their spinal cords which is incorrect.

Author Response

Thank you very much for reviewing my manuscript and providing valuable comments for this revision. I would like to reply to your comments as follows:

The review by Kumamoto attempts to compare the electrophysiological actions of oxytocin and orexins within the spinal cord to various other neuromodulators that alter nociception. This is a huge undertaking and there is a lot of good information in this review however I feel the overall message is being let down by the lack of structure.

The anti-nociceptive properties of oxytocin and orexins A and B are paradoxical when their cellular actions are considered. They directly depolarise superficial spinal cord neurons and increase excitatory transmission, all of which would be expected to facilitate nociception. However they also differentially facilitate GABAergic and glycinergic transmission, which would be expected to reduce nociception (i.e. analgesic). Presumably the net anti-nociceptive properties of these neuropeptides are due to the relative distribution of their receptors and perhaps selective preferential coupling to effectors within particular spinal cord neurons/circuits that promote inhibition of nociceptive signalling. If this is the case then the complex circuitry and heterogeneity of spinal cord neurons has to be considered in relation to neuromodulator effect (i.e. does the neuromodulator target specific subsets of cells? Is facilitation of excitatory inputs within a particular spinal cord circuit that actually results in net inhibition of LI projection neurons?).

Response:

Thank you very much for your valuable opinion. As pointed out by you, the antinociceptive properties of hypothalamic neuropeptides (oxytocin and orexins) will be due to the relative distribution of their receptors and selective preferential coupling of effectors within excitatory and inhibitory spinal dorsal neurons/circuits. As a first step to know the neuropeptides’ roles in antinociception in the spinal dorsal horn, their actions on synaptic transmission in lamina II neurons were examined by using the blind whole-cell patch-clamp technique. As a result, it was found out that the neuropeptides directly depolarize lamina II neurons and increase spontaneous excitatory transmission while differentially facilitating GABAergic and glycinergic spontaneous transmission. There will be a particular spinal dorsal horn circuitry that produces a net inhibition of the projection neurons transferring nociceptive information to the brain as a result of synaptic modulation produced by the hypothalamic neuropeptides in the lamina II. Such a circuitry remains to be revealed. This idea has been given in lines 979-982 in the revised manuscript.

In addition, neuropathic or inflammatory pain can drastically alter the levels of endogenous neuromodulator or receptor expression (e.g. see Imlach et al, 2015 Mol Pharmacol, 88 (3), 460-468, PMID 26104547 showing increased adenosine tone and enhanced A1R sensitivity at excitatory synapses). These are important points that govern the action of each neuromodulator but were either not discussed or not clearly defined in the current review and this needs to be addressed.

Response:

Thank you very much for your valuable opinion. According to your instruction, I mentioned an enhancement of adenosine A1 receptor sensitivity at excitatory but not inhibitory synapses in the lamina II in a rat partial nerve-injury model of neuropathetic pain [Imlach et al., 2015] in lines 520-523 in the revised manuscript. Moreover, I mentioned in lines 962-965 in the revised manuscript, “Since neuropathic or inflammatory pain can drastically change the levels of endogenous neuromodulator or the function of receptor for neuromodulator, as seen in the cases of nociceptin, adenosine and galanin, such changes also will have to be taken into consideration in pain neuromodulators’ effects on nociceptive transmission.”

In this regard, it would be helpful for the reader to have a cartoon of the relevant spinal cord circuitry and the various different cell-types and inputs within it to refer to. It would also be helpful if the effects of each neuromodulator were discussed in the context of this circuitry.

Response:

Since there is few information about the actions of oxytocin and orexins in different types of lamina II neurons, I could not give a cartoon of the relevant spinal cord circuitry in the neuropeptides’ actions. It is unknown how a lamina II neuron sensitive to oxytocin or orexins is connected to a projection neuron transferring nociceptive information to the brain, i.e., mono- or polysynaptically through excitatory or inhibitory neurons. Although it has been reported that lamina II neurons exhibiting orexin B activity are greater in proportion in radial or vertical neurons than central or unclassified neurons in the young rat (see lines 309 and 311 in the revised manuscript), it is unknown whether this result is applied to the adult rat and also what role the different-type neurons play in a spinal dorsal horn circuitry involved in pain transmission. Since orexin receptors exist in both postsynaptic neurons and presynaptic neuron terminals, a schematic diagram exhibiting a spinal cord circuitry involved in pain modulation will become to be very complex.

Each of the neuromodulators discussed act at a GPCRs however the author only intermittently indicated which particular G-protein was engaged (i.e. Gi/o, Gs, Gq) and which intracellular pathways were upregulated by each neuromodulator. This needs to be clarified for all neuromodulators discussed. In addition, no mention of bg subunit signalling was alluded to and this is a key component of the direct membrane potential changes resulting from GIRK activation or CaV inhibition following GPCR activation, particularly those coupled to Gi/o. In fact, the strength of receptor coupling to each effector can markedly alter the outcome of receptor activation and this can vary between CNS regions and between neuronal subtypes within a single region. These points need to be discussed.

Response:

Thank you very much for your valuable advice. As instructed by you, when data are available, I has mentioned an involvement of Gi/o, Gs, or Gq/11 protein and also of a or bg subunit for all neuromodulators discussed. Please see red fonts throughout the text in the revised manuscript.

There are a number of incorrect statements or omissions that need to be amended throughout the manuscript:

Page 2 line 48: ‘naïve’ should be changed to ‘sham’

Response:

As instructed by you, “naïve” has been changed to “sham”; please see line 49 in the revised manuscript.

2)      Page 2 line 95 to page 3 line 107: The author indicates TTX has no effect on spontaneous currents recorded in the spinal cord and that sEPSC/sIPSCs are the same as mEPSCs/mIPSCs. This is simply not true. sEPSC/sIPSCs include action-potential driven events (intrinsic or network activity) whilst mEPSC/mIPSCs represent spontaneous release in the absence of AP generation and/or propagation. They are two very different measures and do not equate the same thing. Therefore, the author needs to specify whether sPSC or mPSCs are being referred to throughout the manuscript.

Response:

Thank you very much for your valuable opinion. As instructed by you, I have deleted a sentence, “sPSCs corresponded to mPSCs”; please see the first and second paragraphs on page 3 in the revised manuscript. Moreover, I have specified whether sPSCs or mPSCs are being referred to throughout the manuscript.

3)      Page 8, lines 385-388: reduced opioid binding sites following dorsal rhizotomy or capsaicin treatment does not indicate opioid receptors are localised at both pre- and post-synaptic sites, it just indicates these treatments alter opioid receptor levels. Their subcellular localisation can only be determined either with EM or functionally via the various electrophysiological recordings detailed throughout the paper.

Response:

Thank you very much for your valuable opinion. According to your suggestion, I have changed “in both nerve terminals and postsynaptic neurons” to “in the dorsal horn”; please see line 396 in the revised manuscript.

4)      Page 8 line 406: the author states sEPSC frequency in diminished by DAMGO & DPDPE and this indicated mu and delta opioid receptors are in the terminals of glutamatergic interneurons. Since sEPSCs are a measure of all spontaneous excitatory transmission, it is impossible to determine whether this is originating from interneurons or external inputs such as the primary afferent neurons.

Response:

Thank you very much for your advice. I revised the sentences; please see lines 407-415 in the revised manuscript.

5)      Page 9, line 420-421: the postsynaptic effects of opioids needs expanding. Opioids have a complex and selective action on subpopulations of dorsal horn neurons. For example the mu and delta receptor agonists produce an outward hyperpolarising current in distinct subsets of neurons within LII neurons but in LI neurons both agonists can have an effect in the same neuron.

Response:

Thank you very much for your valuable advice. According to your instruction, I have mentioned that m, d and k-opioid receptor agonists produce a hyperpolarizing effect in distinct subsets of neurons within lamina II neurons by referring a paper of Eckert & Light (2002); please see lines 430-434 in the revised manuscript.

6)      No mention of kappa opioid receptor or deltorphin effect in the spinal cord.

Response:

Thank you very much for your valuable instruction. As instructed by you, I added an effect of kappa opioid receptor agonist in lamina II neurons in lines 416-420 in the revised manuscript.

7)      Page 9, Lines 418 & 449: the author states GABA and glycine transmission is not affected by DAMGO but please see Kerchner & Zhuo (2002), J Neurophysiology, 88(1), 520-522.

Response:

Thank you very much for your valuable opinion. According to your instruction, I have introduced Kerchner & Zhuo (2002)’s data that are different from our result; see lines 438-441 in the revised manuscript.

8)      Page 9, Line 432: “nociceptin peptide and mRNA” is likely a misprint. Since nociceptin is cleaved from a precursor protein (pre-pronociceptin), it does not have dedicated mRNA, rather pre-pronociceptin mRNA levels are used as an indicator of nociceptin transcription.

Response:

Thank you very much for your valuable advice. As instructed by you, I have used “pre-pronociceptin mRNA”; please see line 455 in the revised manuscript.

9)      Several detailed reviews have already been written on the electrophysiological actions of nociceptin and opioids (e.g. see PMIDs: 10998530, 30838458 & 24762001), these could be referenced to strengthen these sections.

Response:

Thank you very much for your valuable instruction. I have cited PMID 10998530 (Moran et al., 2000) in lines 479-482, PMID 30838458 (Winters et al., 2019) in lines 451-454, and PMID 24762001 (Schröder et al., 2014) in lines 451-454 and 964 in the revised manuscript.

10)   The statement “indicating heterogeneity of lamina II neurons” has been used frequently throughout the manuscript (e.g. P11, line 568; p12 line619…). This statement isn’t incorrect – LII neurons are diverse but it can be expanded upon. For example, P11, line 568: the varying response to 5HT (inward vs outward current) indicates heterogeneity in 5HT receptor subtype expression between different LII neurons and indeed Lu & Perl (see PMID 17463043) show an inward/outward current can vary between LII (& LI) neuronal subtypes (central, radial, vertical, islet). Being more specific on this heterogeneity where possible will also address my first comment.

Response:

Thank you very much for your useful advice. I have changed a sentence about the heterogeneity: please see lines 254-255, 308-309, 424, 605-606, 662-663 and 811-812 in the revised manuscript. I have cited “Lu & Perl (2007; PMID 17463043)” in line 606 in the revised manuscript. Since outward and inward currents produced by 5-HT are examined in morphologically-identified neurons, this result has been mentioned in lines 611-615 in the revised manuscript.

11)   The table should include whether the neuromodulators are pro- or anti-nociceptive when then are delivered intrathecally, systemically or I.C.V (this can vary depending on the neuromodulator and delivery method).

Response:

Thank you very much for your valuable advice. I have shown by using asterisk whether the neuromodulator is antinociceptive or pronociceptive; please see Table 1 in the revised manuscript. Its administration method (intrathecally) has been mentioned in the text.

12)   Page 18 line 905: “there are glycinergic but not GABAergic synapses in the adult rat SDH” suggests adult rats have no GABAergic transmission in their spinal cords which is incorrect.

Response:

Thank you very much for your valuable advice. This sentence has been changed to another one, “There are neurons exhibiting synaptically-evoked glycine but not GABA responses in the adult rat SDH”; please see lines 958-959 in the revised manuscript.

Reviewer 4 Report

 This manuscript addresses an important review in cellular mechanisms for antinociceptin produced by oxytocin and orexins in the rat spinal lamina II.  The review will be important for the full understanding of the regulation of pain transmission and the development of therapeutic drugs and its methods.  The review is clearly presented.  I have annotated the manuscript with several minor corrections, which I believe will improve the readability of the paper.

Major concern

1.     It would be useful to include the advantage of oxytocin and orexins for treatment of pain therapy in comparison with various endogenous pain modulators, in section 5.

Minor comments

1.    L48. primary-afferent evoked → primary-afferent-evoked

2.    L50. “difference” ,Please add description of difference on inflammatory pain.

3.    L61. slice preparation → slice preparation. (Please add period. )

4.    L87. Ad-fiber or C-fiber evoked → Ad-fiber- or C-fiber-evoked

5.    L448. Comp B → J113397

Author Response

Thank you very much for reviewing my manuscript and providing valuable comments for this revision. I would like to reply to your comments as follows:

This manuscript addresses an important review in cellular mechanisms for antinociceptin produced by oxytocin and orexins in the rat spinal lamina II.  The review will be important for the full understanding of the regulation of pain transmission and the development of therapeutic drugs and its methods.  The review is clearly presented.  I have annotated the manuscript with several minor corrections, which I believe will improve the readability of the paper.

Major concern

It would be useful to include the advantage of oxytocin and orexins for treatment of pain therapy in comparison with various endogenous pain modulators, in section 5.

Response:

Thank you very much for your valuable opinion. Oxytocin, orexins A and B exclusively activate oxytocin, orexin-1 and orexin-2 receptors, respectively, while many of other antinociceptive modulators activate several receptor subtypes whose activation results in distinct synaptic modulation. Therefore, pain therapy by oxytocin or orexins in the spinal cord level may have an advantage over other antinociceptive modulators in activating only one kind of receptor leading to inhibitory transmission enhancement. This idea has been added in the last paragraph of section 5 in the revised manuscript.

Minor comments

primary-afferent evoked → primary-afferent-evoked

Response:

Such an addition of hyphen has been performed; please see line 49 in the revised manuscript. A similar addition has been also done in other sentences.

“difference” ,Please add description of difference on inflammatory pain.

Response:

I have added a description of difference on inflammatory pain; please see line 51 in the revised manuscript.

slice preparation → slice preparation. (Please add period. )

Response:

I have added ages of rodents from which slice preparations were dissected; please see line 63 in the revised manuscript.

Ad-fiber or C-fiber evoked → Ad-fiber- or C-fiber-evoked

Response:

Such an addition of hyphen has been performed; please see line 90 in the revised manuscript. A similar addition has been also done in other sentences.

Comp B → J113397

Response:

I have given both of their names; please see line 473 in the revised manuscript.

Round 2

Reviewer 1 Report

I was hoping to see a major revision of this manuscript, which has not been done. I provided specific examples to show how some sections were incomplete, misleading or (strangely) showed methods and results (which I did not expect in a review). These were meant to be just be examples, rather than the only issues that needed to be addressed. 

An example of this are the sections that describe data (eg: line 127 onwards) which is presented as though this the first time these results have been shown. I am still confused about why these are included in a review. This is just one of many issues I have with this draft.

I do really believe that this review needs to revised properly - this should be the responsibility of the author rather than reviewer.

Author Response

Thank you very much for reviewing my revised manuscript. I would like to reply to your comments as follows:

I was hoping to see a major revision of this manuscript, which has not been done. I provided specific examples to show how some sections were incomplete, misleading or (strangely) showed methods and results (which I did not expect in a review). These were meant to be just be examples, rather than the only issues that needed to be addressed.

An example of this are the sections that describe data (eg: line 127 onwards) which is presented as though this the first time these results have been shown. I am still confused about why these are included in a review. This is just one of many issues I have with this draft.

I do really believe that this review needs to revised properly - this should be the responsibility of the author rather than reviewer.

Response:

Recently, we examined the actions of hypothalamic neuropeptides, oxytocin and orexins, on synaptic transmission in lamina II neurons of adult rat spinal cord slices by using the blind whole-cell patch-clamp technique with the aim to know their antinociceptive mechanisms at the synapse level. As a result, we found out that there is a similarity or difference among the neuropeptides and other pain modulators in synaptic modulation in adult rat lamina II neurons. Since there are no reports other than ours about the actions of oxytocin and orexins in adult rat lamina II neurons, at first I introduced their experimental data. Then, these data were compared with those of other pain modulators that were previously reported by using the same preparation and technique, as shown in Table 1. This was why I described such data (eg: line 127 onwards) in this review article.

I have corrected sections that may have been incomplete and misleading; this correction has been shown by red fonts throughout the text.

Reviewer 2 Report

I recommend publication

Author Response

I recommend publication.

Reply to comments:

Thank you very much for reviewing my manuscript.

Reviewer 3 Report

The author has made significant changes to the manuscript which have addressed most of my previous major concerns. However, I still think that the overall structure of the manuscript could be improved upon. The abstract and introduction do not help the reader understand the purpose of the review. i.e. to compare orexin and oxytocin action in the spinal cord with other neuromodulators, which only really becomes clear in the last section. Comparing so many neuromodulators is a huge undertaking and at times it is difficult to follow and in some instances quite light in detail. I would have prefered more detailed information on fewer neuromodulators. Nevertheless, in its present state, I believe this manuscript would be a good baseline resource for anyone interested in neuromodulator action in the spinal cord. 

corrections still needed:

Line 75-76: The author indicates GABA and glycine primarily originate from interneurons within the spincal cord. This is not true, spinal dorsal horn neurons also recieve GABA/glycine inputs that originate from supraspinal regions (e.g. RVM). Further, LIII neurons which provide a large glycinergic input would not technically be decribed as interneurons. In addition, LII contains glutamatergic interneurons.

Author Response

Thank you very much for reviewing my revised manuscript. I would like to reply to your comments as follows:

Comments and Suggestions for Authors

The author has made significant changes to the manuscript which have addressed most of my previous major concerns. However, I still think that the overall structure of the manuscript could be improved upon. The abstract and introduction do not help the reader understand the purpose of the review. i.e. to compare orexin and oxytocin action in the spinal cord with other neuromodulators, which only really becomes clear in the last section. Comparing so many neuromodulators is a huge undertaking and at times it is difficult to follow and in some instances quite light in detail. I would have prefered more detailed information on fewer neuromodulators. Nevertheless, in its present state, I believe this manuscript would be a good baseline resource for anyone interested in neuromodulator action in the spinal cord.

Response:

Thank you very much for your valuable suggestion. According to your advice, I have revised the abstract and introduction; please see lines 25-28 on page 1 in Abstract and lines 69-75 on page 2 in Introduction in the re-revised manuscript.

corrections still needed:

Line 75-76: The author indicates GABA and glycine primarily originate from interneurons within the spincal cord. This is not true, spinal dorsal horn neurons also recieve GABA/glycine inputs that originate from supraspinal regions (e.g. RVM). Further, LIII neurons which provide a large glycinergic input would not technically be decribed as interneurons. In addition, LII contains glutamatergic interneurons.

Response:

Thank you very much for your valuable comments. As instructed by you, I have added a description about GABA/glycine inputs that originate from supraspinal regions such as RVM while deleting a word “mainly”. For this addition, I have given two references that support this fact. I have changed “interneuron” to “neuron”. I have already mentioned “glutamatergic interneurons in the lamina II”. Please see lines 81-85 on page 2, line 572 on page 11 and lines 869 and 880 on page 17 in the re-revised manuscript.

Round 3

Reviewer 1 Report

The revised manuscript has a few changes that address some of the reviewers concerns, but it still needs more work before it is ready for publication. My major concerns are that it still feels like a mix between an original research paper and a review and the lack of understanding of a TTX resistant PSC.

Many sections sound like the results being presented are new findings, yet the authors have previously published this work.  Their justification in the response to reviewers comments is “Since there are no reports other than ours about the actions of oxytocin and orexins in adult rat lamina II neurons, at first I introduced their experimental data.” I think that all of this detail can come out and the main points summarised with references to the original papers. The level of detail describing experimental conditions (eg: how many minutes oxytocin is superfused etc) is not necessary for a review paper and its makes it very hard to read and very confusing. If this data is going to be described in this way, then the relevant figures need to be included as well, or the authors could just refer to their published work and get rid of all the detail.

The authors are still very confused about spontaneous and miniature PSPs. Their statements about TTX having no effect on sIPSCs and sEPSCs are extremely misleading and just not true as what they are describing are mPSCS. Reviewer 3 explains this in their comments, but it seems that this has been ignored. This needs to be fixed.

Spelling mistakes have been introduced since the last version, including ‘neuropathetic’ which should be neuropathic.

Author Response

Thank you very much for reviewing my re-revised manuscript. I would like to reply to your comments as follows:

The revised manuscript has a few changes that address some of the reviewers concerns, but it still needs more work before it is ready for publication. My major concerns are that it still feels like a mix between an original research paper and a review and the lack of understanding of a TTX resistant PSC.

Many sections sound like the results being presented are new findings, yet the authors have previously published this work.  Their justification in the response to reviewers comments is “Since there are no reports other than ours about the actions of oxytocin and orexins in adult rat lamina II neurons, at first I introduced their experimental data.” I think that all of this detail can come out and the main points summarised with references to the original papers. The level of detail describing experimental conditions (eg: how many minutes oxytocin is superfused etc) is not necessary for a review paper and its makes it very hard to read and very confusing. If this data is going to be described in this way, then the relevant figures need to be included as well, or the authors could just refer to their published work and get rid of all the detail.

Response:

Thank you very much for your valuable suggestion. According to your instruction, I have deleted sentences about drug superfusion and a detail of drug effect reversibility in presenting data about oxytocin and orexins actions; please see lines 137, 141-142, 192, 210, 213-214, 253, 261, 272-273, 282, 285, 288, 302, 312, 327-328, 340, 343 and 349 in the re-re-revised manuscript. Throughout the text, I have deleted almost all of “superfusing”, “superfused”, “superfusion” and “bath-applied”. Please see lines 405, 419, 461, 502, 534, 568, 603, 755 and 878 in the re-re-revised manuscript.

The authors are still very confused about spontaneous and miniature PSPs. Their statements about TTX having no effect on sIPSCs and sEPSCs are extremely misleading and just not true as what they are describing are mPSCS. Reviewer 3 explains this in their comments, but it seems that this has been ignored. This needs to be fixed.

Response:

As pointed out by you, TTX generally reduces the amplitude and frequency of sPSCs (sIPSCs and sEPSCs) and thus sPSCs are not different from mPSCs in the presence of TTX. However, in our spinal cord slice preparations, TTX did not affect sPSCs, probably because of deafferentiation in the slices used. For instance, please see figures of our previous papers, Fig. 3Cb and Fig. 7Cb in Wang et al., 2018 (Neuroscience 383, 114-128), Fig. 2Ab, Fig. 3C and Fig. 4C in Wang et al., 2018 (BBRC 501, 100-105), Fig. 3c, d in Jiang et al., 2016 (J Neurochem 136, 764-777), Fig. 3a in Zhu et al., 2016 (NeuroReport 27, 166-171), Fig. 2B and Fig. 5B in Jiang et al., 2016 (Biochem Biophys Rep 7, 206-213), Fig. 3A in Xu et al., 2015 (Neurosci Lett 606, 94-99), Fig. 2C, D in Luo et al., 2014 (Brain Res 1592, 44-54), Fig. 2B1, Fig. 5D and Fig. 8A, B in Jiang et al., 2014 (J Neurophysiol 111, 991-1007), Fig. 3Ab in Yue et al., 2013 (J Neurophysiol 110, 658-671), Fig. 3B in Liu et al., 2013 (Mol Pain 9, 16), Fig. 3B in Inoue et al., 2012 (Neuroscience 210, 4013-415), Fig. 5B in Jiang et al., 2009 (Neuroscience 164, 1833-1844), Fig. 6Aa in Fujita et al., 2009 (J Neurophysiol 102, 312-319), Fig. 2A in Yue et al., 2005 (Neuroscience 135, 485-495) and Fig. 4B in Liu et al., 2004 (Brain Res Bull 64, 75-83).

I did not ignore Reviewer 3’s opinion. I had deleted a sentence, “sPSCs corresponded to mPSCs” in my previous manuscript. Moreover, according to Reviewer 3’s instruction, I had specified whether sPSCs or mPSCs are being referred to throughout the manuscript. For example, please see lines 407-408 on page 8, lines 437-438 on page 9, and lines 645-646 on page 13 in the re-re-revised manuscript.

Spelling mistakes have been introduced since the last version, including ‘neuropathetic’ which should be neuropathic.

Response:

Thank you very much for your pointing out the spelling mistake. I have corrected the spelling mistake; please see lines 520 on page 10 in the re-re-revised manuscript. This re-re-revised version has been very carefully checked in English once again.

Round 4

Reviewer 1 Report

I still have a problem with the authors saying that sEPSCs are TTX resistant. This goes against all other literature in the field, except for their own.